# Ventromedial prefrontal neurons represent self-states shaped by vicarious fear in male mice

Ziyan Huang [1,2,3], Myung Chung [1,2,3], Kentaro Tao [1], Akiyuki Watarai[1], Mu-Yun Wang [1], Hiroh Ito [1,2] & Teruhiro Okuyama [1,2] ✉

Perception of fear induced by others in danger elicits complex vicarious fear responses and behavioral outputs. In rodents, observing a conspecific receive aversive stimuli leads to escape and freezing behavior. It remains unclear how these behavioral self-states in response to others in fear are neurophysiologically represented. Here, we assess such representations in the ventromedial prefrontal cortex (vmPFC), an essential site for empathy, in an observational fear (OF) paradigm in male mice. We classify the observer mouse's stereotypic behaviors during OF using a machine-learning approach. Optogenetic inhibition of the vmPFC specifically disrupts OF-induced escape behavior. In vivo $Ca^{2+}$ imaging reveals that vmPFC neural populations represent intermingled information of other- and self-states. Distinct subpopulations are activated and suppressed by others' fear responses, simultaneously representing self-freezing states. This mixed selectivity requires inputs from the anterior cingulate cortex and the basolateral amygdala to regulate OF-induced escape behavior.

For wild social animals, the perception of fear expressed by others is critical for detecting unseen threats[1]. The fear signals from the demonstrator lead the observer to adopt a variety of defensive behaviors, and these complex and flexible responses include not only freezing but also escape behavior. Notably, because the freeze-and-flee response is mutually exclusive, animals should appropriately select the behavioral output and achieve fast switching between these behaviors, as required for effective and versatile responses[2]. The observational fear (OF) paradigm is commonly used to investigate the transmission of fear and its underlying neural mechanisms across species under laboratory conditions[3–5]. In rodents, the freezing behavior of an observer while a demonstrator receives repetitive foot shocks has generally been measured to quantify the vicarious fear response[6–10]; however, the observer's escape behavior as an OF response has only been described in a limited number of studies[7].

Evidence from human functional magnetic resonance imaging (fMRI)[11,12] implied the involvement of the anterior cingulate cortex (ACC) and basolateral amygdala (BLA) in emotional contagion, which is supported by a series of rodent studies showing the indispensable role of these regions in acquiring vicarious fear[6,8–10]. Subsequently, BLA-projecting ACC neurons preferentially encode socially derived aversive cue information[8] to elicit freezing in the vicarious fear response[13]. Consistently, another study reported OF-induced neural activation in the ACC and BLA, with increased amplitude and slowed decay of ACC-to-BLA NMDAR-mediated currents[14].

Previous studies have also indicated a significant role of the medial prefrontal cortex in the cognitive side of empathy[11,15]. Patients with damage to the ventromedial prefrontal cortex (vmPFC), encompassing the prelimbic (PL) and infralimbic (IL) cortices in rodents, have impairments in their ability to express empathic emotions[16] and interpret the emotions of others[17,18]. Furthermore, recent rodent studies regarding vmPFC social functions have yielded vmPFC neural coding of social representation[19] as well as the affective state discrimination of other conspecifics[20]. Notably, a meta-analysis of

[1]Laboratory of Behavioral Neuroscience, Institute for Quantitative Biosciences (IQB), The University of Tokyo, Tokyo, Japan. [2]Graduate School of Medicine, The University of Tokyo, Tokyo, Japan. [3]These authors contributed equally: Ziyan Huang, Myung Chung. ✉e-mail: okuyama@iqb.u-tokyo.ac.jp

rodents' emotional contagion confirmed c-fos activation of PL and IL in both mice and rats[21]. In conjunction, these results suggest a possible function of the vmPFC (encompassing the PL and IL but not the ACC) in the OF; however, the function and in vivo physiological features of vmPFC neurons and their neural networks co-operating with the ACC and BLA remain unclear.

In this study, we explored the function and neural representation of vmPFC neurons in the OF. To objectively classify the complex behaviors during OF, we employed DeepLabCut (DLC)[22] with dimension reduction clustering using t-distributed stochastic neighbor embedding (t-SNE)[23] and identified eight types of stereotypic behaviors. Although vmPFC neuronal activities enabled decoding both stereotypic behaviors and freezing, optogenetic inhibition of the vmPFC specifically disrupted OF-induced escape behavior, but not freezing behavior. We identified two distinct neural subpopulations activated and suppressed when observing a demonstrator receiving foot shocks (i.e., other-shock). Neural activities of other-shock activated and suppressed neurons were negatively and positively correlated with self-freezing, respectively, revealing an intermingled neural representation of other- and self-states in vmPFC neurons. The representation of the self-state in the other-shock activated and suppressed neurons required the ACC-vmPFC and BLA-vmPFC neural inputs, respectively. Surprisingly, optogenetic inhibition of either the ACC-vmPFC and BLA-vmPFC resulted in the acceleration of escape behavior. Our study suggests that intermingled population coding in vmPFC neurons represents self-states shaped by the other-state to elicit OF-induced escape behavior.

## Results

### Classification of stereotypic behavioral patterns of the observer mice during the OF task

The OF task in rodents, in which an observer exhibits robust freezing behavior while a demonstrator receives repetitive foot shocks, has been commonly used to investigate neural mechanisms underlying the vicarious fear response[24]. In our paradigm, a demonstrator and an observer freely moved in the context for 5 min (habituation period), and then the demonstrator received a 2 s shock every 10 s (a total of 60 times) for 10 min (conditioning period) (Fig. 1a). Throughout the OF session, the observer exhibited seemingly variable behaviors, with high levels of freezing during the conditioning period (Fig. 1b). In this study, we exploited a series of unsupervised analyses to categorize the observer's behavior into discrete patterns to reveal a mixture of distinct and stereotyped behavior sequences. First, to extract low-dimensional time-series representations of postural dynamics, we defined and extracted 13 body points on an observer's head, trunk, and limbs in recorded video frames (Fig. 1c) using the markerless tracking method DeepLabCut[22] (Supplementary Fig. 1a). We then divided a single behavior session into 10-s bouts (corresponding to a shock frequency) and embedded the 1950 D (13 body points × 2 coordinates × 7.5 frames/s × 10 s) representation into 2 D space using t-SNE[23,25] followed by density-based clustering[26–28], which yielded a total of eight clusters (Fig. 1d–f; Supplementary Fig. 1b) representing distinct behavioral patterns (Fig. 1g). Plotting the coordinates of the center of the back within each bout revealed whether the animal moved or remained stationary (Fig. 1g, right). By inspecting the observer's posture, position, and movement during each 10-s behavior bout, we annotated each behavioral cluster as follows: (I) move left, (II) move right, (III) back, (IV) front, (V) left, (VI) right, (VII) near a demonstrator (D), and (VIII) far from a demonstrator (D) (Fig. 1g, center and left; Supplementary Fig. 1c). The two clusters corresponding to the moving behavior (clusters I and II) exhibited a significantly lower freezing rate than the other six clusters (clusters III, IV, V, VI, VII, and VIII) (Fig. 1h).

Unexpectedly, the obtained behavioral sequence for each mouse varied from one to the other, with little common behavior at specific times (Fig. 1i). This led us to examine behavioral transitions. To test the

length of previous bouts that significantly affected the prediction of the next bout in a behavioral sequence, we constructed a series of Markov chain models ranging from zeroth (based only on the distribution of behavioral clusters) to the fifth order[29]. The results suggested that only the immediately preceding bout strongly predicted the next bout for all behavioral clusters (Supplementary Fig. 2a). Therefore, we performed sequence analysis between two consecutive bouts (i.e., transitions). The rendered square matrix of the transition probability demonstrated that only the repetition of the same behavioral cluster (i.e., clusters III, IV, V, VI, VII, and VIII) was significant during the conditioning period, and those significant repetitions increased compared with the habituation period across individuals (Supplementary Fig. 2b). In the control group, in which the demonstrator mice did not receive foot shocks (no-shock control group), an increased freezing rate of the observer mice and a strong tendency to repeat the same behavioral cluster were not observed (Supplementary Fig. 3), suggesting that the latter feature along with the freezing behavior were attributed not only to the habituation effect to the shock apparatus but also to fear contagion from the demonstrator to the observer.

We conducted a similar analysis using shorter 2-s bouts to isolate the vicarious fear responses further. As mice move and stop continuously, we set a threshold of 50% in the 2-s freezing rate to classify bouts as mobile or immobile. We then performed unsupervised classification and obtained 9 components for each dataset (Fig. 2a, b, Supplementary Fig. 4a, b). Among all the components (Supplementary Fig. 4c, d), three characteristic components were extracted from the mobile dataset, namely (m1) approaching, (m2) leaving, and (m3) moving near D; three from the immobile dataset, (i1) escaped, (i2) gazing-1, and (i3) gazing-2 (Fig. 2c). The freezing rate of m1 and m2 were significantly lower compared to other mobile components (Fig. 2d, Supplementary Table). When the habituation and conditioning period were compared, the proportion of mobile components (m1–m3) decreased, while the immobile components (i1–i3) increased in the conditioning period (Fig. 2e). Among the eight clusters obtained in Fig.1, m1 shared the largest portion of cluster I, m2 of cluster II, m3 of cluster I and VII, i1 of cluster III and VIII, i2 of V, and i3 of VII (Fig. 2f) indicating that the 10-s clusters contained a mixture of different behavior components.

### Impaired escape behavior with the optogenetic inhibition of vmPFC neurons

To investigate the function of the vmPFC of the observer in the OF, we optogenetically silenced excitatory neuronal activity in the vmPFC during the conditioning period. AAV5-CaMKII-eArchT3.0-eYFP was bilaterally injected and an optic fiber was implanted (Fig. 3a). We confirmed the optogenetic inhibition of the vmPFC using acute single-unit recordings from head-fixed mice (Fig. 3b). First, we found that vmPFC inhibition during the conditioning period did not affect the freezing rate, unlike that of the ACC[6] (Fig. 3c). To take a closer look at the behavioral patterns, we positioned newly obtained data with the vmPFC inhibition on the reference behavior atlas of the mice without any surgery described above (Fig. 1e) by calculating the 10 most correlated reference bouts and placing the new bout at the median coordinate following a previous study[25] (Fig. 3d, e). The transition repertoire exhibited reduced repetition of escape behavior from the demonstrator (i.e., cluster VIII) (Supplementary Fig. 5), corresponding with the low ratio of behavioral cluster VIII compared to the shuffled data during the conditioning period in the inhibition group but not in the control group (Fig. 3g). In line with this, the proportion of the i1 "escaped" was significantly smaller and i2 + i3 "gazing" was significantly larger in the inhibition group compared to the control group (Fig. 3h–j). To further confirm this trend, we investigated the position of the observers. The distance from the demonstrator side was significantly smaller in the

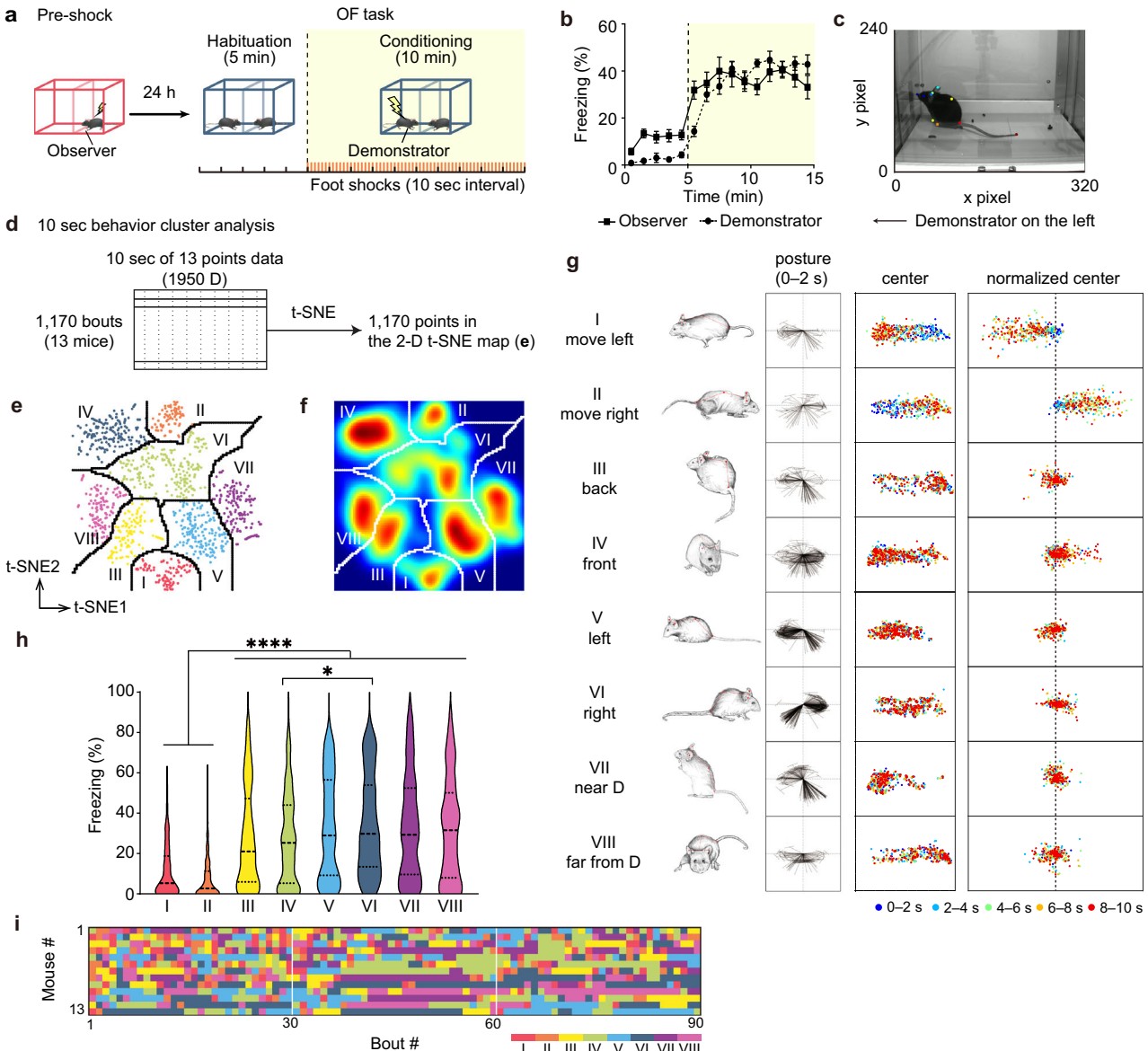

**Fig. 1 | Automatic behavior classification using 10-s bouts of the OF task.**
**a** Behavior paradigm for the OF task. **b** Freezing rate during the OF task ($n = 13$ mice, one-way repeated measures ANOVA). Data are presented as mean ± SEM. **c** Representative image of a frame with 13 body-point tracking. **d** Schematic drawing of the 10-s behavior analysis. **e** Results of t-SNE embedding of 13 body-point tracking experiments using DeepLabCut (data from $n = 13$ mice, 1170 bouts total). **f** Watershed clustering of the t-SNE result. **g** Left, diagram of the skeleton in the first 2 s of 10-s bouts using four points (left ear, right ear, back center, tail root) with the back center point aligned to (0,0) (xpixel: −80 to 80, ypixel: −80 to 80). Center and right, the position of the back center in the chamber (center, xpixel: 0 to 320, ypixel: 0 to 240) and back center points with the position of the first frame of each bout (75 frames) were set to (0,0) (right, xpixel: −280 to 280, ypixel: 0 to 240). **h** Violin plot of freezing rate for each behavioral cluster (one-way ANOVA, Tukey-Kramer test, two-sided). Mean ± SEM. Cluster I: 11.0 ± 1.4%, II 7.3 ± 1.2%, III 27.6 ± 2.2%, IV 26.9 ± 1.5%, V 32.7 ± 1.9%, VI 34.9 ± 1.8%, VII 33.1 ± 2.1%, and VIII 32.1 ± 2.1%. The numbers under each cluster number represent the number of bouts for each cluster. **i** The behavioral sequence of each mouse sorted by the number of transitions. See Supplementary Information for exact $p$ values. Source data are provided as a Source Data file.

inhibition group (Fig. 3k). Together, vmPFC inhibition during OF disturbed the escape behavior from the threat, but not the freezing behavior itself induced by emotional contagion.

**vmPFC neural representation of the stereotypic behavioral patterns and freezing during OF**
To reveal the in vivo physiological properties of vmPFC neurons during the OF task, we performed microendoscopic Ca²⁺ imaging by microinjection of AAV5 carrying the calcium indicator protein GCaMP6f into the vmPFC (Fig. 4a). After signal processing of the calcium fluorescence videos, we identified 355 vmPFC cells from four mice and extracted calcium traces from individual cells (Fig. 4b, c; see Methods).

The behavioral patterns of the recorded mice were identified in the same way as in the optogenetic inhibition experiments described above (Fig. 3d, e), and then the behavioral sequence of each mouse was obtained (Fig. 4d). To investigate how many neurons were specifically activated in mice showing each behavioral pattern among the eight, we statistically identified vmPFC neurons showing a significantly higher number of calcium events in each cluster compared to the event timing shuffling data (see Methods). Consequently, we found that 11.5% of cells (41 of 355) were significantly activated in individual clusters (Fig. 4e), while 9.5% of cells (36 of 377) were identified as cluster-specific neurons in the no-shock control group (Supplementary Fig. 6a, b).

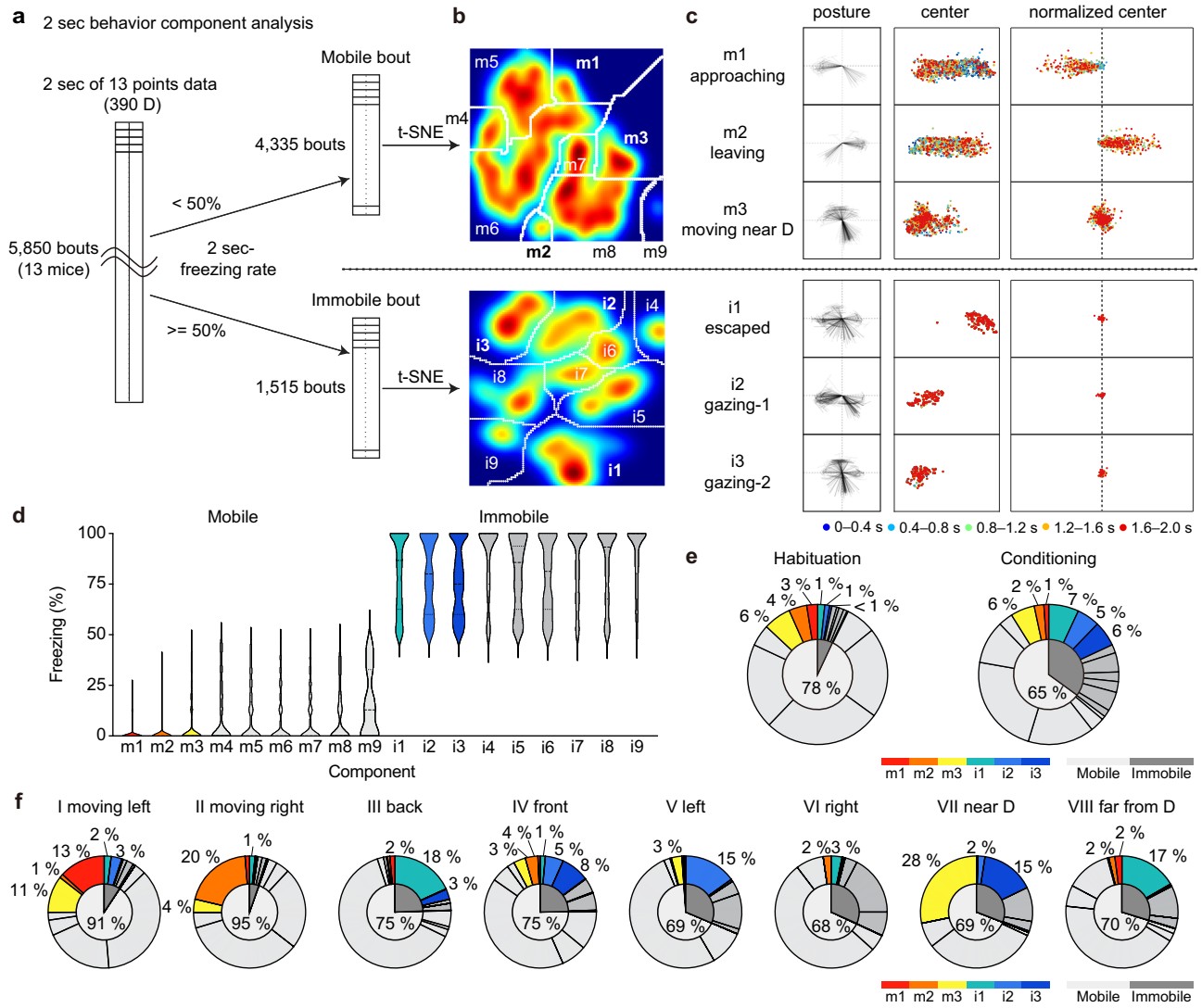

**Fig. 2 | Automatic behavior classification using 2-s bouts of the OF task.** **a** Schematic drawing of the 2-s behavior analysis. **b** Results of t-SNE embedding of 13 body-point tracking experiments using DeepLabCut (data from $n = 13$ mice, mobile bouts: 4335 bouts, immobile bouts: 1515 bouts). **c**, Left, diagram of the skeleton in the 2-s bouts using four points (left ear, right ear, back center, tail root) with the back center point aligned to (0,0) (xpixel: −80 to 80, ypixel: −80 to 80). Center and right, the position of the back center in the chamber (center, xpixel: 0 to 320, ypixel: 0 to 240) and back center points with the position of the first frame of each bout (75 frames) were set to (0,0) (right, xpixel: −280 to 280, ypixel: 0 to

240). **d** Violin plot of freezing rate for each behavioral cluster (one-way ANOVA, Tukey-Kramer test, two-sided). Mean ± SEM. Component m1: 0.9 ± 0.4%, m2 1.5 ± 0.5%, m3 7.8 ± 0.7%, m4 10.4 ± 1.0%, m5 10.9 ± 0.5%, m6 10.2 ± 0.4%, m7 9.9 ± 0.4%, m8 10.4 ± 0.9%, m9 15.9 ± 3.0%, i1 80.4 ± 1.1%, i2 78.2 ± 1.2%, i3 77.0 ± 1.1%, i4 87.4 ± 2.0%, i5 79.9 ± 1.3%, i6 79.0 ± 1.6%, i7 85.9 ± 1.7%, i8 84.0 ± 1.3%, and i9 88.4 ± 1.7%. **e** The proportion of each component during the habituation and conditioning periods. **f** The proportion of each 2-s component in each 10-s cluster. Source data are provided as a Source Data file.

Next, to examine whether the neural population as a whole in the vmPFC represents the behavioral patterns, we utilized a multiclass error-correcting output code model with a support vector machine (SVM) binary decoder. SVM decoders, which were trained to distinguish among the eight types of clusters based on the pattern of vmPFC population activities for each mouse (Fig. 4f, g), showed significantly better accuracy compared to the decoders trained using shuffled data, with an accuracy of 24.9−43.6% (Fig. 4h, i; Supplementary Fig. 7a; see Methods). In the no-shock control group, the decoders trained with real data did not perform better than those trained with shuffled data (Supplementary Fig. 6c, d). Therefore, the vmPFC neural population encodes information about behavioral clusters during the OF task.

We further investigated the 2-s behavior component information encoding in the vmPFC. Since two components, i1 and i2 + i3, showed significant decrease and increase, respectively, in the vmPFC inhibition group compared to the control group (Fig. 3), we hypothesized that

information regarding these behavior components is particularly represented in vmPFC neurons. We found 17 of 355 neurons significantly active at the component i1 and 11 of 355 at the component i2 + i3 with 2 overlapping neurons (Fig. 4j).

We also tested whether the vmPFC neural population represented freezing behavior, another aspect of behavioral output during OF. The performance of SVM decoders for the binarized freezing states (high or low, binarization threshold is the median freezing value for each mouse) from the neural population activity was significant, with an accuracy of 68.0−75.9% (Fig. 4k−n; Supplementary Fig. 7b; see Methods). In the no-shock control group, there was no significant difference between the decoders trained with the shuffled and real data (Supplementary Fig. 6e, f). Taken together, these results suggest that the vmPFC population neurophysiologically represents multifaceted behavioral states of the self, including both escape and freezing behavior, which were specifically induced by OF.

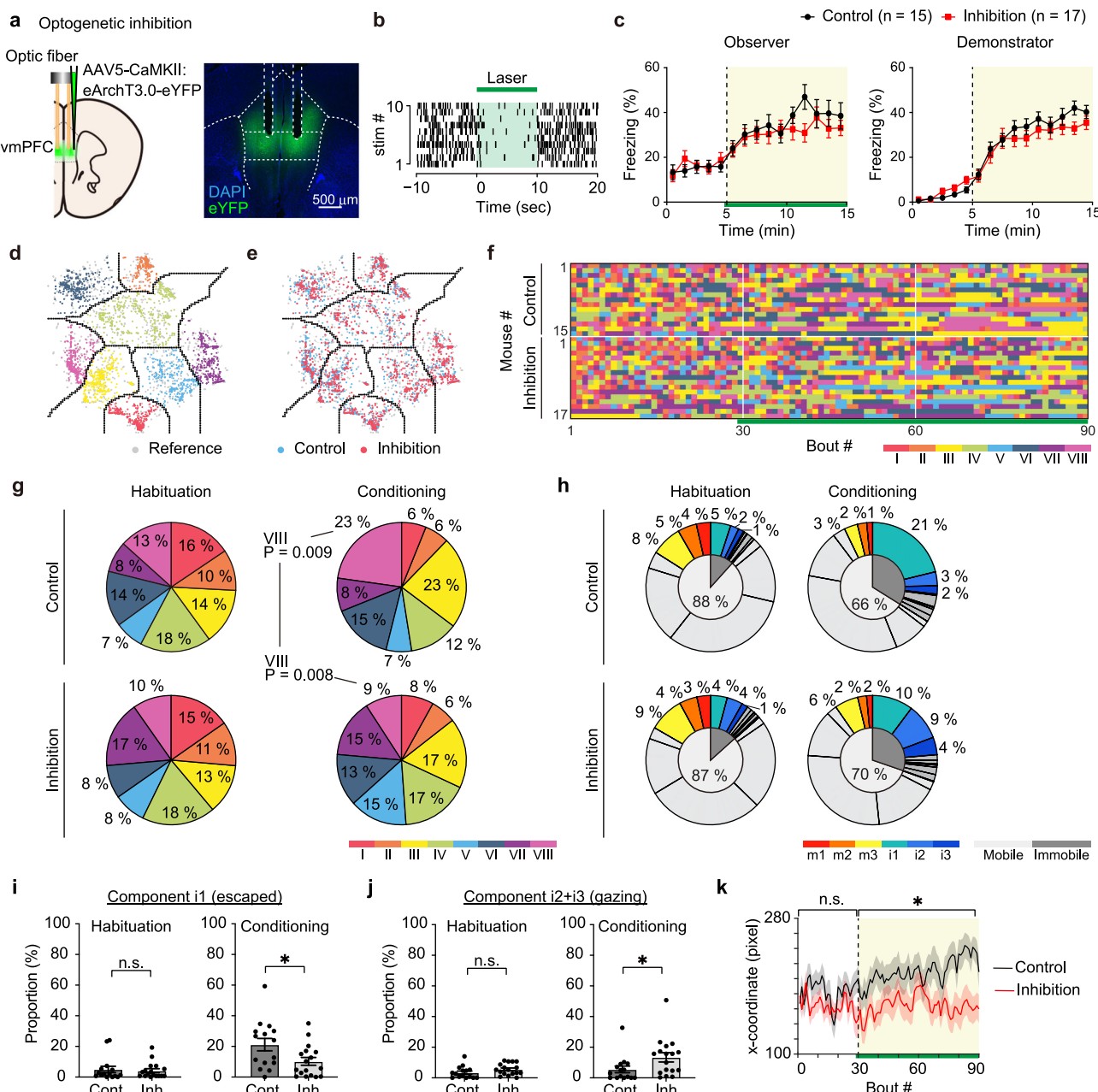

**Fig. 3 | Optogenetic inhibition of the vmPFC during the OF task. a** Left, schematic representation of the optogenetic manipulation of the vmPFC. Right, coronal vmPFC section image stained with anti-GFP (green, for eYFP) and DAPI (4′,6-diamidino-2- phenylindole, blue). **b** Representative in vivo unit recording of optogenetic inhibition. **c** Freezing rate of the observer and the demonstrator (two-way repeated-measures ANOVA). Green line indicates the light manipulation. **d**, **e** Positioning points of the control (n = 15 mice, light blue dots) and inhibition (n = 17 mice, pink dots) data on the t-SNE atlas obtained from the previous experiment (Fig. 1d, gray dots). **f** Behavioral sequence of each mouse during the OF task sorted by the number of transitions within each group. **g** Proportion of each behavioral cluster in the habituation and conditioning periods (the permutation

test, two-sided with Bonferroni correction, P < 0.025/8 = 0.0031; STAR Methods). **h** Proportion of each behavioral component in the habituation and conditioning periods. **i** Proportion of components i1 of the control and inhibition group in the habituation and conditioning periods (control: n = 15 mice, inhibition: n = 17 mice, unpaired t-test, two-sided). **j** Proportion of components i2 + i3 of the control and inhibition group in the habituation and conditioning periods (control: n = 15 mice, inhibition: n = 17 mice, unpaired t-test, two-sided). **k** The distance from the demonstrator side. The x-coordinate of the back center of the control and inhibition groups (two-way repeated-measures ANOVA). *P < 0.05, n.s. not significant. Data are presented as mean ± SEM (error bars and shadows). See Supplementary Information for exact p values. Source data are provided as a Source Data file.

## Other-shock and self-freezing are correlated inversely in the vmPFC

Next, we investigated how the vmPFC neural representation of self-states is associated with other-state, namely, other-shock information. Sensory modalities, including vision and sound, convey information about fear expressed by the demonstrator to the observer[6]. During the OF task, the demonstrator responded differently: jumped and ran around with screaming vocalization while the shock was delivered

(0–2 s) and mostly froze during the shock interval (2–10 s)[13]. We observed periodic features in the observer's behavioral changes during each shock frequency (10 s) (Supplementary Fig. 8). Freezing rate was significantly lower at the shock timing (0–2 s, Supplementary Fig. 8a), although the average movement speed along the x-axis had no temporal feature (Supplementary Fig. 8b). The behavioral features indicate that the observer's behavior is dependent on demonstrator being shocked. Thus, to investigate the effect of other-shock

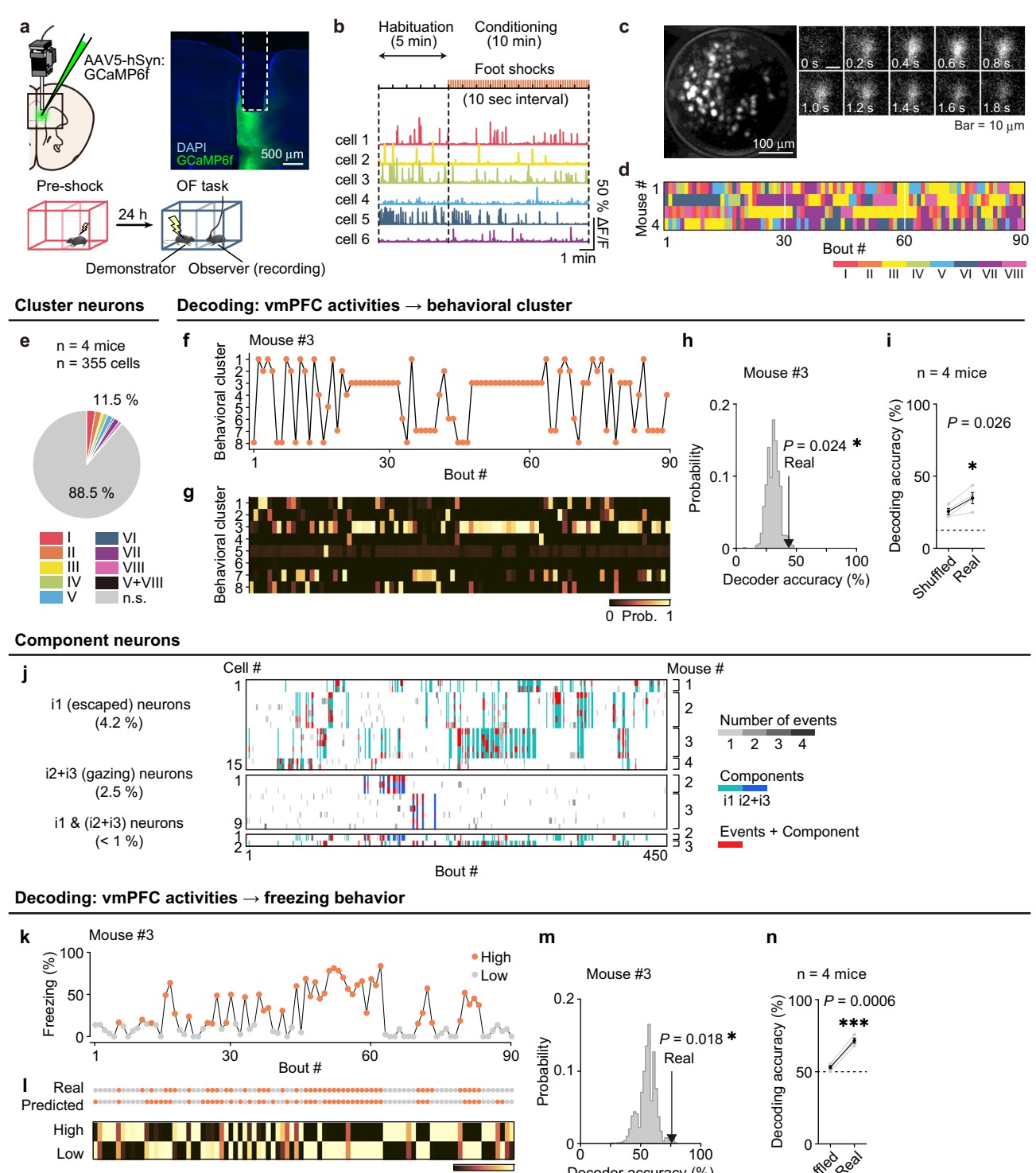

**Fig. 4 | Behavioral cluster and freezing encoding in the vmPFC. a** Left, schematic illustration of the microendoscopy of the vmPFC. Right, coronal vmPFC section image stained with anti-GFP (green, for GCaMP6f) and DAPI (blue). Bottom, recording paradigm. **b** Representative calcium transients. **c** Left, representative stacked image acquired during a 15-min microendoscopy recording. Right, time-lapse image sequence of GCaMP6f fluorescence in a single cell. **d** Behavioral sequence of each mouse (*n* = 4 mice). **e** Proportion of cluster-specific neurons (*n* = 4 mice, *n* = 355 cells). **f** Behavioral cluster sequence of mouse #3. **g** Decoding results of the behavioral cluster sequence of mouse #3. **h** Histogram of the decoder accuracy calculated by the shuffled data of mouse #3. The arrow indicates the accuracy and one-sided p-value of the real data. **i** Decoding accuracy of the

behavioral cluster compared to the shuffled data (*n* = 4 mice, paired *t*-test). **j** Firing patterns of the i1 "escaped" component-specific neurons, i2 + i3 "gazing" component-specific neurons, and i1&i2 + i3 component-specific neurons. **k** Raw freezing rate with the binarized freezing states of mouse #3. **l** Decoding results of the freezing states of mouse #3. **m** Histogram of the decoder accuracy calculated by the shuffled data of mouse #3. The arrow indicates the accuracy and one-sided *p*-value of the real data. **n** Decoding accuracy of the binarized freezing states compared to the shuffled data (*n* = 4 mice, paired *t*-test). *P < 0.05, ***P < 0.001, n.s. not significant. Data are presented as mean ± SEM. See Supplementary Information for exact *p* values. Source data are provided as a Source Data file.

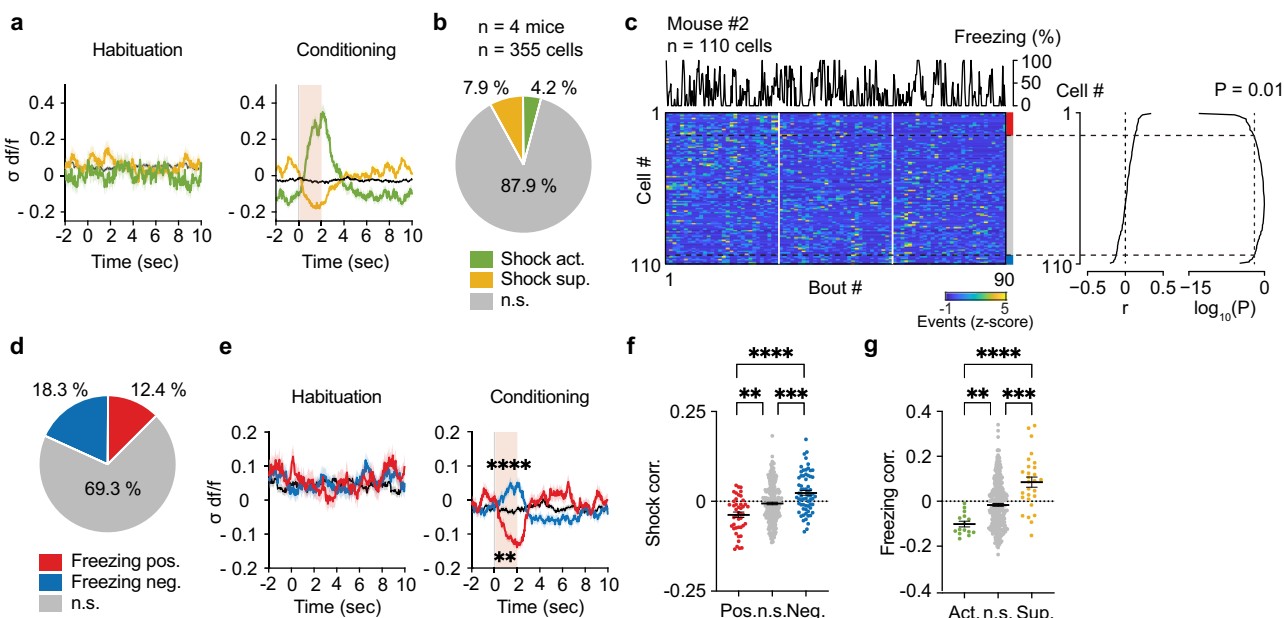

**Fig. 5 | Other-shock and self-freezing encoding in the vmPFC. a** Change in activity of other-shock responding neurons during 2-s shock bouts compared to preceding 2-s interval. The black line represents the n.s. group. **b** Proportion of other-shock responding neurons. **c** Neural activity sorted by the degree of correlation with the self-freezing rate of mouse #2 (Pearson's correlation coefficient, two-sided). **d** Proportion of neurons significantly correlated with self-freezing rate. **e** Change in activity of self-freezing correlated neurons during 2-s shock bouts compared to preceding 2-s interval (Wilcoxon signed-rank test, two-sided; −2–0 s vs 0–2 s). The black line represents the n.s. group. **f** Shock correlations of self-freezing correlated neurons (Kruskal–Wallis test, $n = 355$ cells). **g** Freezing correlations of other-shock responding neurons (Kruskal–Wallis test, $n = 351$ cells). **$P < 0.01$, ***$P < 0.001$, ****$P < 0.0001$, n.s. not significant. Data are presented as mean ± SEM (error bars and shadows). See Supplementary Information for exact $p$ values. Source data are provided as a Source Data file.

responses on vmPFC neuronal activities, we first compared the activity difference of each neuron between their responses before and during the shock moment (−2–0 s. vs. 0–2 s). Fifteen of 355 (4.2%) recorded neurons were identified as shock-activated neurons, and 28 of 355 (7.9%) as shock-suppressed neurons (Fig. 5a, b; see Methods). Both shock-activated and shock-suppressed neurons did not show periodic calcium transients every 10 s during the habituation period (Fig. 5a), suggesting that the activity alterations at the shock onset were induced by OF.

We also cogitated how, at the neuronal level, this other-shock information could be linked to the self-freezing information. Among the wide range of correlations between neuronal activity and freezing rate (Fig. 5c; see Methods), we identified a sizable fraction of neurons as either positively (44 of 355, 12.4%) or negatively (65 of 355, 18.3%) correlated with the observer's freezing (Fig. 5d). Strikingly, freezing positively- or negatively-correlated neurons showed significantly suppressed or activated calcium transients during the shock moment (0–2 s) compared to the baseline (Fig. 5e), and these changes in flow during the last 10 min were not observed in the no-shock control group (Supplementary Fig. 6g, h). Correspondingly, the neurons that were positively or negatively correlated with the observer's freezing rate showed a significantly negative or positive correlation with the shock-or-not states (1 or 0) (Fig. 5f). In contrast, shock-activated neurons showed a significantly negative correlation with freezing rate, while shock-suppressed neurons showed a positive correlation (Fig. 5g). Notably, while the vmPFC has cellular diversity, with each sub-population of neurons targeting different major downstream regions[30,31], the spatial distribution of these other-shock responding and self-freezing correlated neurons did not segregate on the imaging plane (Supplementary Fig. 9a–i). While there were some overlaps between shock-responding neurons and freezing-correlated neurons, not all neurons exhibited both representations (Supplementary Fig. 9j). Taken together, the information on other-shock and self-freezing is represented in the vmPFC via two distinct neuronal subpopulations with opposite responses: freezing-positive/shock-suppressed neurons and freezing-negative/shock-activated neurons.

Furthermore, we investigated the neural properties of i1 "escaped"-specific and i2 + i3 "gazing"-specific neurons, which are major components of the freezing-positively correlated neurons and shock-suppressed neurons (Supplementary Fig. 10a–d). The calcium activity of these neurons was suppressed during the shock timing (Supplementary Fig. 10e, f), and their correlations with the freezing rate were significantly higher than that of non-significant neurons (Supplementary Fig. 10g, i). Also, while not significant, shock correlations showed lower trends than non-significant neurons (Supplementary Fig. 10h, j). Notably, we confirmed that the activity of the i1-specific neurons was high at the timing of component i1 compared to the preceding and proceeding 2-s bouts and component i2 + i3 compared to the proceeding 2-s bouts (Supplementary Fig. 10k, l).

## Physiological activity in the vmPFC during inhibition of the ACC-vmPFC and BLA-vmPFC pathways

Several studies have shown that the acquisition of vicarious fear and subsequent expression of freezing behavior highly depends on neural activity in the ACC[6,8,9] and BLA[8,14]. Given that vmPFC neurons receive direct neural projections from both the ACC and BLA[32–35], we conjectured whether the vmPFC received OF-related other-shock or self-freezing information via projections from these brain regions. To address this hypothesis, the vmPFC calcium transients were recorded with the AAV5-hSyn:GCaMP6f injection into the vmPFC, while inhibiting its input from the ACC or BLA by injecting AAV5-CaMKII:NpHR3.0-mCherry into each region and applying laser stimulation for silencing in the vmPFC during the conditioning period (ACC-vmPFC circuit inhibition, Fig. 6a–h; BLA-vmPFC circuit inhibition, Fig. 6i–p). For ACC-vmPFC inhibition (384 cells from four mice), other-shock responding neurons and self-freezing correlated neurons were identified (shock-activated, 17 of 384 cells, 4.4%; shock-suppressed, 30 of 384 cells, 7.8%; freezing positively correlated, 45 of 384 cells, 11.7%; freezing negatively

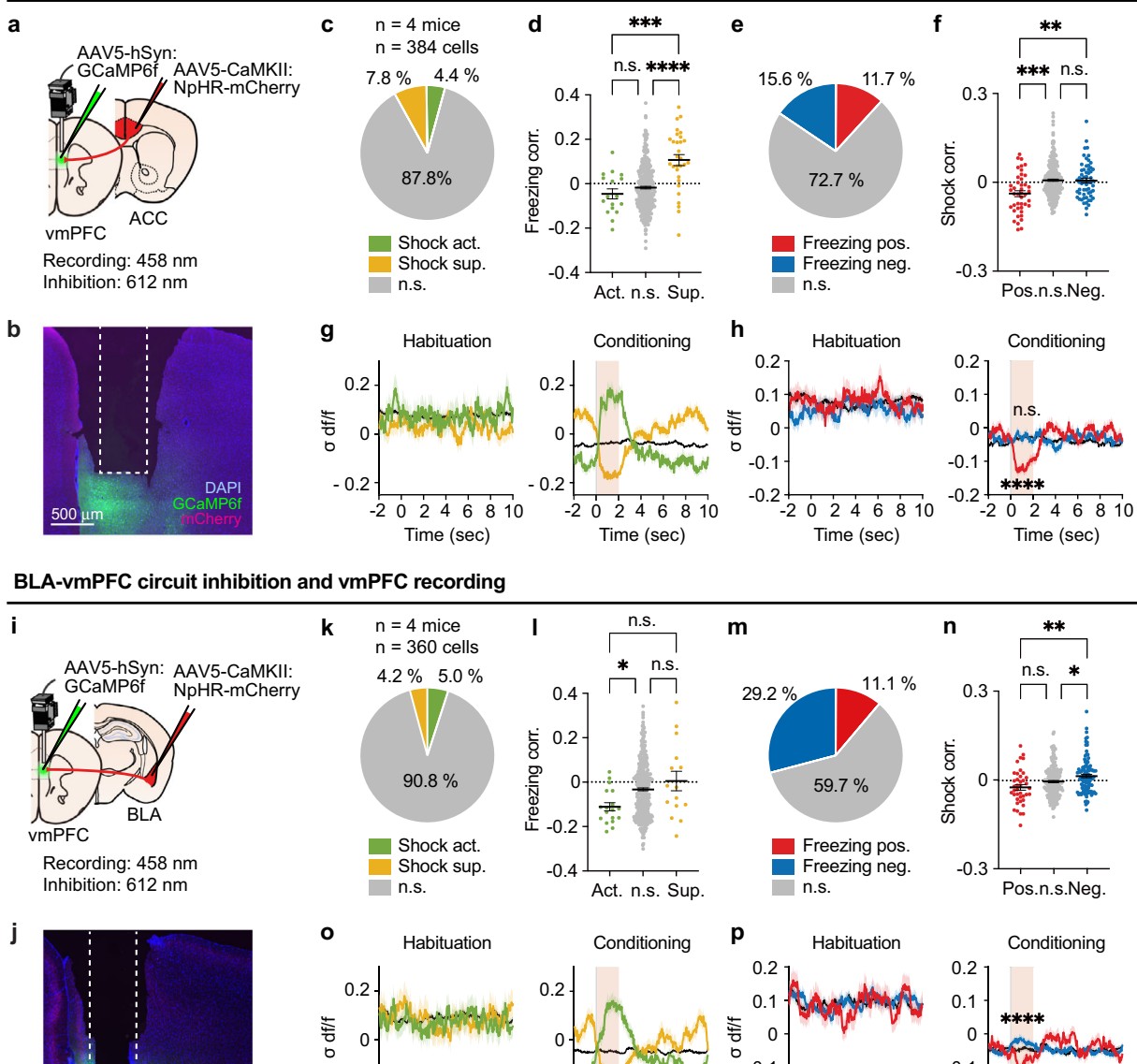

**Fig. 6 | Other-shock and self-freezing link derived by ACC-vmPFC and BLA-vmPFC input. a** Schematic illustration of the microendoscopy of the vmPFC with ACC-vmPFC optogenetic inhibition. **b** Coronal GRIN lens implanted vmPFC section image stained with anti-GFP (green, for GCaMP6f), anti-RFP (magenta, for mCherry) and DAPI (blue). **c** Proportion of other-shock responding neurons with ACC-vmPFC inhibition. **d** Freezing correlations of other-shock responding neurons with ACC-vmPFC inhibition (Kruskal–Wallis test, $n = 384$ cells). **e** Proportion of neurons significantly correlated with the freezing rate with ACC-vmPFC inhibition. **f** Shock correlations of self-freezing correlated neurons with ACC-vmPFC inhibition (Kruskal–Wallis test, $n = 384$ cells). **g** Change in activity of other-shock responding neurons with ACC-vmPFC inhibition during 2-s shock bouts compared to preceding 2-s interval. The black line represents the n.s. group. **h** Change in activity of self-freezing correlated neurons with ACC-vmPFC inhibition during 2-s shock bouts compared to preceding 2-s interval (Wilcoxon signed-rank test, two-sided; −2–0 s vs 0–2 s). The black line represents the n.s. group. **i** Schematic illustration of the microendoscopy of the vmPFC with BLA-vmPFC optogenetic inhibition. **j** Coronal

GRIN lens implanted vmPFC section image stained with anti-GFP (green, for GCaMP6f), anti-RFP (magenta, for mCherry) and DAPI (blue). **k** Proportion of other-shock responding neurons with BLA-vmPFC inhibition. **l** Freezing correlations of other-shock responding neurons with BLA-vmPFC inhibition. (Kruskal–Wallis test, $n = 356$ cells). **m** Proportion of neurons significantly correlated with the freezing rate with BLA-vmPFC inhibition. **n** Shock correlations of self-freezing correlated neurons with BLA-vmPFC inhibition (Kruskal–Wallis test, $n = 360$ cells). **o** Change in activity of other-shock responding neurons with BLA-vmPFC inhibition during 2-s shock bouts compared to preceding 2-s interval. The black line represents the n.s. group. **p** Change in activity of self-freezing correlated neurons with BLA-vmPFC inhibition during 2-s shock bouts compared to preceding 2-s interval (Wilcoxon signed-rank test, two-sided; −2–0 s vs 0–2 s). The black line represents the n.s. group. *$P < 0.05$, **$P < 0.01$, ***$P < 0.001$, ****$P < 0.0001$, n.s. not significant. Data are presented as mean ± SEM (error bars and shadows). See Supplementary Information for exact $p$ values. Source data are provided as a Source Data file.

correlated, 60 of 384 cells, 15.6%) (Fig. 6c, e; Supplementary Fig. 11a). Interestingly, the freezing correlation of shock-activated neurons was not significantly different from that of non-significant neurons (Fig. 6d, green), unlike the mice without pathway inhibition (Fig. 5g, green), whereas that of shock-suppressed neurons remained positive (Figs. 5g, 6d, g, yellow). Conversely, the freezing negatively correlated neurons did not show a significantly positive correlation with the shock-or-not states (Fig. 6f, blue), which was observed in mice without pathway inhibition (Fig. 5f, blue), whereas freezing positively correlated neurons (Figs. 5f, 6f, red). Corresponding with these results, the increased calcium transients of the freezing negatively correlated neurons at the shock moment (0–2 s) were not observed, whereas the decreased transients of the positively correlated neurons were maintained (Fig. 6h). Therefore, the mixed selectivity of the freezing-negative/shock-activated subpopulation required input from the ACC.

Moreover, a distinct pattern of effects was seen for the BLA-vmPFC inhibition mice (360 cells from four mice). The freezing correlation of shock-suppressed neurons (18 of 360 cells, 5.0%) (Fig. 6k) was not significantly different from that of the non-significant neurons (Fig. 6l, yellow), unlike the mice without pathway inhibition (Fig. 5g, yellow), while shock-activated neurons (15 of 360 cells, 4.2%) showed a negative correlation with the freezing rate (Figs. 5g, 6l,o, green). Conversely, freezing positively correlated neurons (40 of 360 cells, 11.1%) (Fig. 6m; Supplementary Fig. 11d) were not significantly negatively correlated with the shock-or-not states (Fig. 6n, red), unlike the mice without pathway inhibition (Fig. 5f, red), in contrast, to negatively correlated neurons that maintained their positive correlation with the shock-or-not states (Figs. 5f, 6n, blue). Notably, both the suppressed responses at the shock moment (0–2 s) of the freezing positively correlated neurons and the activated responses of the negatively correlated neurons (105 of 360 cells, 29.2%) were still detected (Fig. 6p), probably because of the unilateral partial inhibition of the vmPFC. Together, these results clearly showed that both the ACC-vmPFC and BLA-vmPFC circuits play essential roles in linking the neural representation of other-shock and self-freezing during OF. Importantly, the ACC-vmPFC and BLA-vmPFC have opposite functions; the former contributed to the activities of the freezing-negative/shock-activated neurons, and the latter contributed to the freezing-positive/shock-suppressed neurons, suggesting the cooperative function of two distinct pathways for processing the neural representations of self-states based on other-state.

### Optogenetic inhibition of ACC-vmPFC and BLA-vmPFC accelerates the escape behavior

To further investigate whether the ACC-vmPFC and BLA-vmPFC pathways also facilitate behavioral outputs during OF, especially escape behavior that required vmPFC function (Fig. 3), we performed an optogenetic inhibition of each pathway. We injected AAV9-CaM-KII:NpHR3.0-eYFP into the ACC (Fig. 7a, b) or BLA (Fig. 7d, e) and performed axon terminal inhibition in the vmPFC. The freezing rate did not change in either case (Fig. 7c, f). While the proportion of i1"escaped" tends to be larger and i2 + i3 "gazing" smaller in the inhibition group compared to the control group, these differences did not reach statistical significance (Fig. 7g, h). To describe the difference in detail, we rigorously quantified escape behavior by calculating how much more time each mouse spent on the near-side to the demonstrator (near-side) or the far-side from the demonstrator (far-side) during the conditioning period than during the habituation period (Fig. 7i; see Methods). Thus, we found that the inhibition accelerated the observers to move further away (toward the furthest far-side) from the demonstrator side in both experiments (Fig. 7j, k). Together, these results suggest that both of the two distinct neural subpopulations in the vmPFC regulate the escape behavior during OF.

## Discussion

In this study, we showed the neural representation of the vmPFC of other- and self-states during OF, which is critical for modulating OF-induced escape behavior. Here, we primarily discuss the (1) neural mechanisms underlying OF-induced escape behavior and (2) vmPFC neural representation in the OF on self-states and other-states.

While previous studies regarding the OF task have focused mainly on freezing behavior, very few studies have focused on other defensive behaviors, including escape behavior[7]. In our study, following the objective classification algorithm that was successfully applied to zebrafish behaviors and fly songs in past research[26,27], a t-SNE-based clustering analysis using body-point data tracked by DeepLabCut automatically classified eight types of stereotypic 10-s behavior clusters and 2-s behavior components during the OF task that could not be segmented by the freezing rate alone (Figs. 1h, 2d). Eight behavior clusters consisted of different characteristic behavior components (Fig. 2f). Furthermore, we confirmed that these eight types of behavior were detected robustly, even when using a subset of the acquired dataset (Supplementary Fig. 1b). Our results revealed that observers exhibited an increase in the repetition of the same behavioral cluster during the conditioning period (Supplementary Fig. 2b), suggesting that mode switching of the self-state depends on the other-state at the behavioral level.

Furthermore, optogenetic inhibition of the CaMKII-expressing vmPFC neurons (i.e., excitatory neurons) impaired escape behavior from the demonstrator side (Fig. 3g–k). In contrast, while the differences in the proportion of components were small, the inhibition of the ACC and BLA neural inputs to the vmPFC accelerated the escape behavior, implying the OF-induced escape behavior is regulated via an inhibitory interneuron-mediated neural network (Fig. 7l). In rodents, the vmPFC, both PL and IL, is a critical region for the regulation of active avoidance in non-social Pavlovian fear conditioning, whereas its inactivation does not affect the freezing evoked by a conditioned stimulus[36–39]. In addition, the dmPFC, encompassing the PL and ACC, and its neural input from the BLA are also crucial for the initiation of avoidance behavior evoked by threat-predicting stimuli[39], suggesting that the mPFC is involved in dynamic mode switching from freezing to escape behavior under exposure to aversive stimuli. Additionally, considering our results together, the vmPFC dysfunction presumably disrupted the initiation of escape behavior in the OF.

Regarding the intermingling neural representations of self-freezing and other-shock in the vmPFC, we obtained three following insights. First, our in vivo physiological recordings of vmPFC neurons with SVM decoders showed that vmPFC neural populations represent other-state-dependent self-state information regarding variable behavioral outputs, such as our identified stereotypic behaviors and freezing (Fig. 4i, n). These neural representations were only depicted in the mice under the OF task, whereas the mice without observing other-shock had no decodable representation, although cluster-specific neurons and freezing-correlated neurons were identified (Supplementary Fig. 6b, g). Notably, considering the high similarity of vmPFC neural population activities maintained up to 12 s during the OF task, 10 s of the behavior bout length (used as it corresponded to the shock frequency) was an adequate period (Supplementary Fig. 7c, d). Interestingly, although optogenetic inhibition of the vmPFC did not impair OF-induced freezing behavior (Fig. 3c), some vmPFC neurons exhibited neuronal activities significantly correlated with the freezing rate, corresponding with high decoding accuracy for freezing from vmPFC population activities, suggesting that the vmPFC possesses significant information on self-freezing. In the case of stereotypic behaviors, each behavioral output reflected by the self-states can be decoded from population activities, even when a single vmPFC neuron selectivity to that aspect is eliminated. Although the decoding accuracy of behavioral clusters was higher than that of the random data, the overall rate remained below 50%. This could be because the 10-s bouts contained a

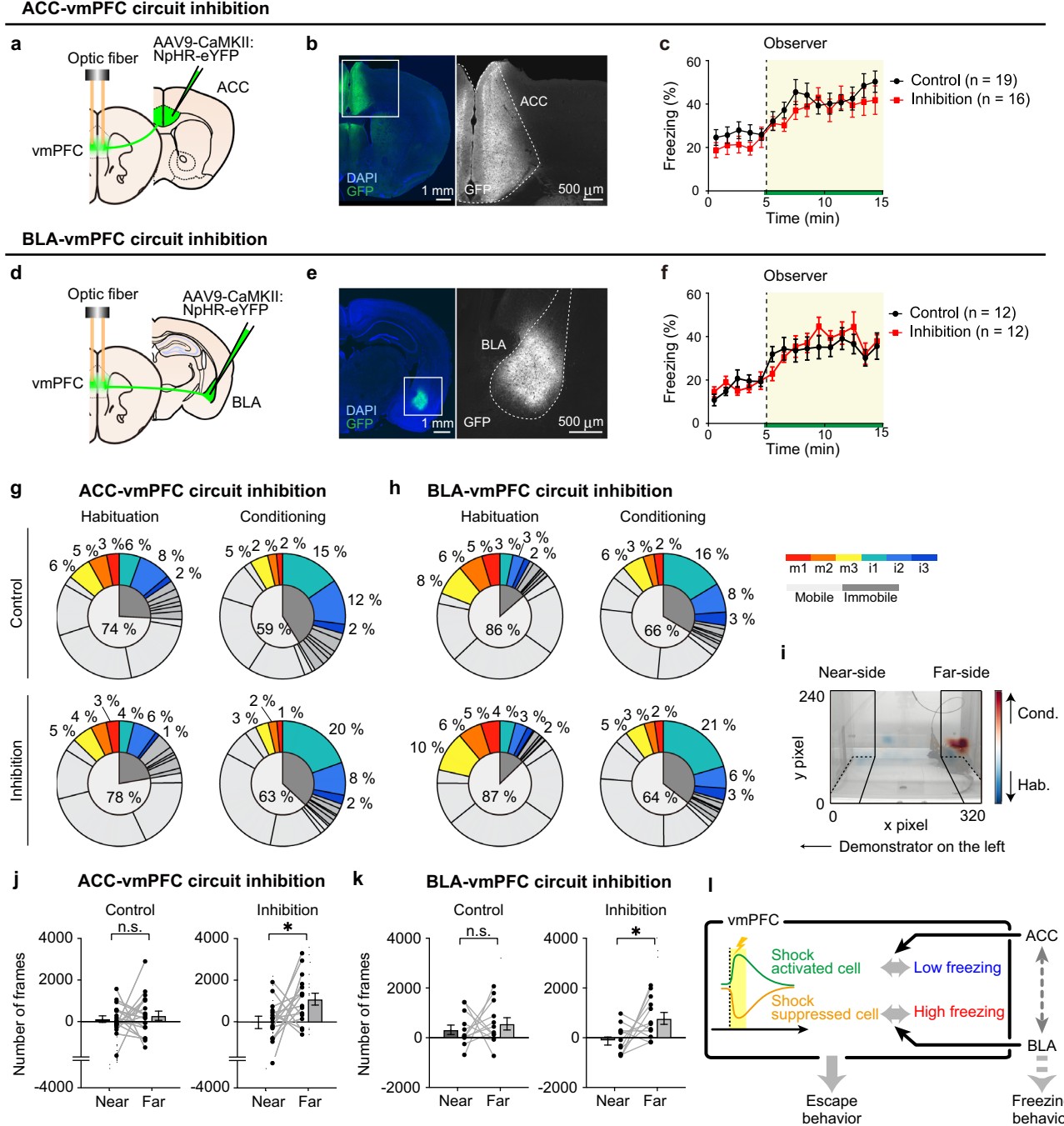

**Fig. 7 | Optogenetic inhibition of the ACC-vmPFC and BLA-vmPFC circuits.**
**a** Schematic illustration of the ACC-vmPFC optogenetic inhibition. **b** Coronal vmPFC section image stained with anti-GFP (green, for eYFP) and DAPI (blue). **c** Freezing rate of the observer during the OF task with ACC-vmPFC inhibition. The green line indicates the light manipulation. **d** Schematic illustration of the BLA-vmPFC optogenetic inhibition. **e** Coronal vmPFC section image stained with anti-GFP (green, for eYFP) and DAPI (blue). **f** Freezing rate of the observer during the OF task with BLA-vmPFC inhibition. The green line indicates the light manipulation. **g** Proportion of each behavioral component in the habituation and conditioning periods with ACC-vmPFC circuit inhibition. **h** Proportion of each behavioral

component in the habituation and conditioning periods with BLA-vmPFC circuit inhibition. **i** Representative image and heatmap of the back center position, the conditioning period position count − 2 × the habituation period position count. **j** The number of frames on the near-side and the far-side from the demonstrator with ACC-vmPFC inhibition (control: $n = 19$ mice, inhibition: $n = 16$ mice, paired $t$-test). **k** The number of frames on the near-side and the far-side from the demonstrator with BLA-vmPFC circuit inhibition (control: $n = 12$ mice, inhibition: $n = 12$ mice, paired $t$-test). **l** Graphical model of the suggested mechanism of the vmPFC. *$P < 0.05$, n.s. not significant. Data are presented as mean ± SEM. See Supplementary Information for exact $p$ values. Source data are provided as a Source Data file.

mix of components, albeit with varying proportions (Fig. 2f). However, our analysis did reveal the presence of neurons that were specifically active during the escaped (i1) and gazing (i2 + i3) behaviors (Fig. 4j), which are major components of freezing positively-correlated neurons (Supplementary Fig. 10a). We thus speculate that these distinct

components underlie the significantly higher decoding rate for eight clusters compared to the shuffled data. Although it would be unreliable to decode those behavior components with the unbalanced dataset in its current form, such analysis may provide more evidence to support this hypothesis with a larger dataset. These representations

of self-states probably rely on two possible neural encoding mechanisms: the distributed and coordinated activities of neural populations[40,41], and the coding based on different activity characteristics, such as the spike intervals[42].

Second, we identified a link between information about other-shock and self-freezing within vmPFC neurons. We identified two distinct neuronal subpopulations in the vmPFC with opposite responses, the activities of which simultaneously represent other-shock and self-freezing. The neural subpopulation was activated by other-shock, and its activity was negatively correlated with the self-freezing rate, and vice versa. Previous studies have also found that neural populations in the ACC[8] and BLA[13] were activated and suppressed by other-shock (or its associated cue). Moreover, in the rat ACC, emotional mirror-like neurons exist that have the capability to respond to both self-fear and other-fear in different contexts[9]. In contrast, we here found that the vmPFC is the region in which single-neuron activity is tuned to the mixed and simultaneous representation of both other-state and self-state information during OF, suggesting the mixed selectivity of vmPFC neurons in the social context. Theoretical studies have shown that mixed selectivity has a significant computational advantage for the repertoire of input-output functions implementable by readout neurons[40]. The high dimensionality of the vmPFC might contribute to the remarkable adaptability of neural coding and variable defensive behavioral outputs in the social context.

Third, optogenetic inhibition of the ACC-vmPFC and BLA-vmPFC projections disrupted the link between other-shock and self-freezing within two distinct neuronal subpopulations: freezing-negative/shock-activated neurons and freezing-positive/shock-suppressed neurons, respectively. These two pathways function in parallel and cooperatively for the high dimensionality of the vmPFC representation, presumably to facilitate variable vicarious fear responses. By optogenetic inhibition, the mixed selectivity of the two neural subpopulations representing both the other-state and self-state disappeared (i.e., the pure selectivity representing either other-state or self-state), and the OF-induced escape behavior was impaired as well. In the current working model for the OF, the ACC-BLA network is the center for eliciting OF-induced freezing behavior[6,8,13,43]. Together with previous studies, our study proposes two parallel information networks in the OF: the ACC-BLA network for freezing; and the ACC-vmPFC and BLA-vmPFC networks for escape behavior.

The essential role of the BLA-mPFC circuits in the self-fear response has been extensively studied using the fear conditioning paradigm. The PL of the vmPFC has been implicated in sustained fear expression and resistance to fear memory extinction, whereas the IL plays a central role in the extinction[44]. The basal amygdala in the BLA is the major source of fear-related input to both the PL and the IL[35,45,46]. Another optogenetic study revealed that the BLA-vmPFC facilitates anxiogenic behavior and reduces social interaction[47]. Our study shows the significant function of the BLA-vmPFC in the vicarious fear response.

Converging evidence from humans, nonhuman primates, and rodents has proposed the PFC as a central hub of the social brain[48–50] for multi-layered social cognition such as empathy[51,52] social hierarchy[53,54], social reward monitoring and valuation[55], social decision-making[49], and high-level social cognition in humans (i.e., perspective-taking and mentalization)[17,56–58]. All these social cognitive processes require other- and/or self-representation and its integration. Our study sheds light on the role of the vmPFC in the neural representation of self-states shaped by others' states.

## Methods

### Animals

All procedures were performed in accordance with protocols approved by the Institutional Animal Care and Use Committee at the Institute for Quantitative Biosciences, the University of Tokyo (Protocol number 2915 (2018), 3112 (2019), 0201 (2020), 0314 (2021), 0403-2 (2022), A2022IQB018 (2023)). All mice described in this paper were C57BL/6 J (B6) male mice (Clea Japan). All animals were housed in the Institute for Quantitative Biosciences facility under a 12 h (7 am–7 pm) light/dark cycle, 23 ± 2 °C, 50% humidity with food and water *ad libitum*.

### Optogenetics and microendoscopy surgery

Mice were anesthetized with mixed anesthetics (0.75 mg/kg medetomidine, 4.0 mg/kg midazolam, and 5.0 mg/kg butorphanol) and mounted on a stereotaxic apparatus (Leica Angle Two, Leica Biosystems). A glass pipette (World Precision Instruments) attached to a 1 ml microsyringe (Hamilton) filled with mineral oil was used for the microinjection of virus vector solutions using a microsyringe pump (UMP3, World Precision Instruments) to control the injection speed and volume. The glass pipette was slowly lowered to the target site, and the virus solutions were delivered at a speed of 2–3 nl/s and then retracted 5 min after injection.

For the optogenetic inhibition experiments, bilateral virus delivery and optic fiber implants were aimed at coordinates relative to Bregma: vmPFC injections were targeted to +1.80 mm Anterior-Posterior (AP), ±0.35 mm Medial-Lateral (ML), and −2.50 mm Dorsal-Ventral (DV); ACC injections were targeted to +1.00 mm AP, ±0.25 mm ML, and −2.10 mm DV; BLA injections were targeted to +1.60 mm AP, ±3.30 mm ML, and −4.85 mm DV; and vmPFC optic fiber implants (Ø200 μm core, 0.22 NA, Doric Lenses Inc.) were placed at +1.80 mm AP, ±0.35 mm ML, and −2.35 mm DV. For the vmPFC optogenetic inhibition experiments, 150 nl of rAAV5/CaMKIIa:eArchT3.0-eYFP (University of North Carolina [UNC] Vector Core, $3.4 \times 10^{12}$ virus molecules/ml, inhibition) or rAAV5/CaMKIIa:eYFP (UNC Vector Core, $3.6 \times 10^{12}$ virus molecules/ml, control) was injected into the vmPFC. In axon terminal optogenetic inhibition experiments, 120 nl and 150 nl of rAAV9/CaMKIIa:eNpHR3.0-eYFP (Addgene, $2.2 \times 10^{13}$ virus molecules/ml, adjusted to $2.2 \times 10^{12}$ virus molecules/ml, inhibition) or rAAV9/CaMKIIa:eGFP (University of Pennsylvania Vector Core, $2.5 \times 10^{13}$ virus molecules/ml, adjusted to $5.0 \times 10^{11-12}$ virus molecules/ml, control) was injected into the ACC and BLA, respectively.

For microendoscopic Ca²⁺ imaging, 300 nl of rAAV5/hSyn:G-CaMP6f (UNC Vector Core) was unilaterally injected into the vmPFC. For the Ca²⁺ imaging experiment combined with circuitry optogenetic inhibition (ACC-vmPFC and BLA-vmPFC), 120 nl and 150 nl of rAAV5/CaMKIIa:eNpHR3.0-mCherry (UNC, $4.7 \times 10^{12}$ virus molecules/ml) were bilaterally injected into the ACC and the BLA, respectively. Immediately following virus injection, a 500 μm gradient index (GRIN) lens (Doric Lenses Inc.) was implanted 150 μm above the injection site (+1.80 mm AP, +0.35 mm ML, −2.35 mm DV). The mice were allowed to recover for 2–4 weeks before the behavioral experiments.

### Acute single-unit recordings from head-fixed mice and optogenetic inhibition

Two adult male mice (3–4 months old) were anesthetized with 1–2% isoflurane and mounted on a stereotaxic apparatus (SR-9M-HT, Narishige). After the virus was microinjected into the bilateral vmPFC (as described above), a custom-made head frame was attached to the cleared skull using dental cement for subsequent recording sessions. After 1–2 weeks of recovery, the mice were anesthetized and mounted with a head frame, and a craniotomy was performed around the virus injection site. A 64-channel silicon probe (A4 × 16-Poly2-5 mm-20s-150-160, NeuroNexus) conjugated with a fiber optic cannula (Ø105 μm core, 0.22 NA; CFMLC21U-20, Thorlabs) was attached on a micromanipulator and was gradually inserted into the target area (+1.60–2.05 mm AP, +0.35 mm ML, −2.00–2.50 mm DV) perpendicular to the midline. The probe was connected to a 64-channel amplifier board (RHD2164, Intan Technologies), and neural activity was sampled at 30 kHz using the Open Ephys data acquisition system[59]. The fiber optic cannula was connected to a yellow-green diode-pumped solid-state laser (MGL-FN-561-100mW, CNI Laser) through a fiber patch

cable. After the neural signal was stabilized, a pulse train (10 s duration, 20 s interval, 10 pulses) was generated using PulsePal (Sanworks)[60], and optogenetic inhibition (5 mW at the fiber tip; approximately a half of laser intensity used in the OF task because of unilateral inhibition) was delivered to the recording site. The recorded spike signals were automatically sorted using Kilosort2[61] followed by manual adjustment of the waveform clusters using the Phy graphical user interface[62].

## Histology and immunohistochemistry
Mice were transcardially perfused with 4% paraformaldehyde (PFA) in phosphate-buffered saline (PBS) for post-hoc analysis. Extracted brains were fixed in 4% PFA solution overnight and coronally sectioned to a thickness of 50 μm using a vibratome (VT1000S, Leica), with every three sections collected. Sections were incubated in 0.3% Triton-X PBS with 5% normal goat serum (Vector Laboratories) for 1 h at room temperature (RT), followed by the addition of a primary antibody to the solution and incubation overnight at 4 °C. The primary antibodies used were chicken anti-GFP (Thermo Fisher Scientific, A10262, 1:1000) for eYFP, eGFP, and GCaMP6f staining, and rabbit anti-RFP (Rockland, 600-401-379, 1:1000) for mCherry staining. After rinsing four times with PBS, brain sections were incubated with secondary antibodies, anti-chicken Alexa Fluor-488 conjugated secondary antibodies (Thermo Fisher Scientific, A11039, 1:500) and anti-rabbit Alexa Fluor-546 conjugated secondary antibodies (Thermo Fisher Scientific, A11010, 1:500), in 0.3% Triton-X PBS with 5% normal goat serum for 3 h at RT. Sections were rinsed twice with PBS and then stained with DAPI (4′,6-diamidino-2- phenylindole, 1 μg/ml) dissolved in PBS, and rinsed again with PBS. The brain sections were then mounted onto glass slides with VectorShield (Vector Laboratories) or Fluoromount/Plus (Diagnostic BioSystems) medium. Images were obtained by fluorescence microscopy (BZ-X710, Keyence) using 4× and 10× objectives or confocal laser microscopy (FV3000, Olympus) using 10× and 20× objectives. Images were post-processed using the ImageJ software (NIH).

## Observational fear (OF) task
All habituation and behavior assays were conducted during the facility's light cycle (7 am–7 pm). The same mice were never used twice in the same behavioral paradigm. Twelve- to 20-week-old male mice were used as observers and demonstrators. The observer mice were socially housed, whereas the demonstrator mice were singly or socially housed. The mice that underwent optic fiber, or GRIN lens implantation surgery, were co-housed immediately after the surgery. An observer and a non-littermate demonstrator mouse were co-caged to familiarize each other (starting from day 1)[63,64]. The observer and demonstrator were then habituated to the experimenter's hands for another 2 days (days 4 and 5) for three times in total. Previous research has shown that observers with prior foot shock experience respond more to demonstrators' fear[65]. Based on the results, a shock experience was given to the observer in a different context. Observer mice were placed in the chamber and partitioned using a matt-gray divider in the middle. A 2-s foot shock (0.75 mA) was delivered at 140 s for 3-min pre-shock conditioning. Acetic acid (1%) was used as an olfactory cue in the pre-shock conditioning context (pre-shock, day 6). Twenty-four hours later, the observer and demonstrator mice were individually placed in a chamber separated by a transparent plastic divider in the middle (OF, day 7). A matt-gray acrylic plate on the bottom and 0.25% benzaldehyde were used for observer mice to have a different context. After a 5-minute habituation session, a 2-s foot shock was delivered every 10 s for 10 min to the demonstrator mouse. The observer mice with microendoscopy were habituated to the experimenter's hands and a dummy microscope on three separate days (days 4 to 6) for four times in total, and the pre-shock (day 7) was performed 1 day before the OF (day 8). (Note: Mouse #2 performed OF on day 13.) The shock intensity of the OF was 0.75 mA for optogenetic inhibition experiments and 1.0 mA for the others. We did not observe any

behavioral differences between groups using 0.75 mA and 1.0 mA in our setting (Supplementary Fig. 1d). Behavior was recorded using FreezeFrame (Actimetrics) at 7.5 Hz. Optogenetic inhibition started at 290 s until the end of the experiment. The intensity of the 561 nm green laser (described above) for the bilateral optogenetic inhibition was set between 12.5 mW and 13.0 mW. Calcium transients were recorded at 25 Hz using a twist-on eFocus fluorescence microscope (Doric Lenses Inc.). The LED power for unilateral optogenetic inhibition combined with microendoscopic Ca$^{2+}$ imaging (612 nm) was set to 6.0 mW.

## Freezing rate
The freezing rate was calculated using two methods. One method was with FreezeFrame software (Actimetrics) using the algorithm "freezing" for observer without surgery and all demonstrator data (threshold = 6, bout = 0.25 s), and the other was using the algorithm "SMP · optogenetics" in the data from observer mice that had been subjected to surgery for optogenetic manipulation and microendoscopy; threshold = 1; open/close = 1; pixel threshold = 50 (FreezeFrame4, vmPFC inhibition), 200 (FreezeFrame5, vmPFC inhibition), 1 (FreezeFrame5, circuit inhibition) or 50 (FreezeFrame5, microendoscopy).

## Point tracking using DeepLabCut
DeepLabCut[22] was used to track the motion of the mouse. The videos of OF task with a frequency of 7.5 Hz recorded in FreezeFrame were used, and 13 points (nose, right/left ear, right/left eyes, head top, right/left hand, back center, right/left foot, tail root, and tail tip) were chosen for tracking. For post-processing, a median filter within DeepLabCut and linear completion of points with a probability of <0.999 were performed. In addition, as a geometrical restriction, we set the y coordinate of the tail root to be the same as that of the back center if the former was larger than the latter (the back center was essentially above the tail root in our camera angle of view). Additionally, owing to an update to the FreezeFrame software (from version 4 to version 5) in the middle of the vmPFC inhibition experiment, the recorded video size was doubled in both height and width (x:640 pixels, y:480 pixels). Therefore, the tracking results for videos obtained in version 5 were divided by two to obtain the same frame size as the first set of videos (x:320 pixels, y:240 pixels). The speed of each mouse during the OF task was calculated by summing the differences in the x coordinates of the back center of each frame during a 2-s bout.

## t-SNE and clustering
t-SNE is a dimension-reduction algorithm that maintains local distances while ignoring far distances and is suitable for visualization and clustering[23]. Using the data obtained in behavioral tracking using DeepLabCut, we performed t-SNE (MATLAB function: tsne) (1) every 10 s as one bout, as shocks were delivered to the demonstrator mice every 10 s (MATLAB version 2020a), and (2) every 2 s as one bout (MATLAB version 2022a). For 10-s bouts, the 15-min tracking data of an individual mouse were parsed into 90 10-s bouts. Because the x- and y-coordinates of the 13 points on the trunk were tracked in the 10-s video at 7.5 Hz (75 frames), each bout consisted of 1950-dimensional data. For 2-s bouts, the 15-minute tracking data of an individual mouse were parsed into 450 2-s bouts, which consisted of 390-dimensional data. We then divided bouts according to their mean freezing rate (threshold: 50%) into mobile and immobile datasets and performed t-SNE in each dataset. This was followed by clustering using a watershed algorithm (MATLAB function: watershed) for the mouse data (Figs. 1e, f, 2b), as previously described in MATLAB (MathWorks Inc.). We used the Barnes-Hut implementation and set perplexity = 30 as the default for t-SNE embedding. Principal component analysis (PCA) before t-SNE embedding was omitted because the results of t-SNE performed with the PCA-extracted components and the raw data did not show qualitative differences.

### Drawing of posture sequences and body positions

Data from five points (left ear, right ear, ear center, back center, and tail root; ear center is the middle point of the left and right ears) were collected every 2 s in the same frame at the same transparency. The back centers were placed at (0,0). The back center points were pointed in the x- and y-coordinates in the "center", and the points in the first frame of every bout (75 frames) were set to (0,0) in the "normalized center".

### Markov chain modeling

Following a previous study[29], we modeled the eight behavioral clusters as eight states in a Markov process. A zeroth-order model uses only the distribution of clusters for the prediction. The corresponding number of preceding bouts was considered in the higher-order Markov models. A paired two-tailed t-test was performed to compare the prediction accuracy between the two models (e.g., a zeroth-order model vs. a first-order model) with Bonferroni correction ($p < 0.025/5 = 0.005$) for each state.

### The transition from one cluster to another

To identify the significance of each behavioral cluster transition, we used a permutation test and randomly shuffled the behavioral sequence within all mice during the habituation period and the conditioning period 1000 times and calculated two-tailed p-values for each transition.

### Positioning new data on the t-SNE atlas

Although t-SNE is a nonparametric method and no new points can be placed on a t-SNE atlas once constructed, straightforward mapping of new points on an existing t-SNE atlas has been proposed. Following a previous study[25], for each new bout point of optogenetics and microendoscopy data, we calculated the Pearson correlation with every reference bout point of the data of mice without surgery and found its $k = 10$ nearest neighbors. Next, we positioned the point at the median t-SNE location of the k reference points. We then used the borderline obtained in the t-SNE of WT data and attributed clusters to each point (Fig. 3d, e). The procedure was the same for the 10-s bouts and 2-s bouts. We confirmed that clustering based on this alignment provided similarly classified behavioral clusters by illustrating and checking their posture sequence, back center position, and its temporal change.

### The proportion of each cluster

To identify the significance of the proportion of each behavioral cluster, we used a permutation test and randomly shuffled the label of the group (control or inhibition), calculated the proportion of each behavioral cluster in the first 30 bouts (the habituation period) or the last 60 bouts (the conditioning period) 10,000 times, and calculated two-tailed p-values of the real data with Bonferroni correction ($p < 0.025/8 = 0.0031$) for each behavioral cluster. This allowed us to detect a significantly large or small number of bouts for each behavioral cluster between the two groups.

### Side preference analysis

To quantify the side preference of the observer, whether near or far from the demonstrator, two sections were set on the image. The length of the front and back sides of the floor (as the floor was not fully visible in the image, the pseudo-length was calculated by extending the sides of the floor in the image and finding the intersection) were each divided into five equal parts. The section closest to the demonstrator was named the near-side, and the section farthest was named the far-side. The number of frames that the observer stayed in these two compartments was calculated, and the side preference during the conditioning period was compared by subtracting twice the number of

frames during the habituation period (baseline) from the number of frames during the conditioning period. Finally, a paired t-test was performed between the number of frames of the near-side and that of the far-side.

### Calcium events detection

The recorded calcium-imaging movie was aligned and converted to a TIFF file extension using Doric Neuroscience Studio software (Doric Lens Inc.) and then preprocessed with minor modifications. Briefly, the movies were spatially downsampled by two and automatically cropped to extract the round-shaped field of view. Motion correction of the movies was performed using Mosaic MATLAB code (correction type, skew, translation, and rigid; reference region by subtracting spatial mean [$r = 16$ pixels], inverting, and applying spatial mean [$r = 4$ pixels]). The preprocessed movies were then processed using EXTRACT[66] to extract the calcium transients. The automatically suggested cells were then carefully manually selected: A total of 355 cells out of 591 cells from vmPFC (mouse #1, 115 out of 152; mouse #2, 110 out of 179; mouse #3, 100 out of 178; mouse #4, 30 out of 82), 377 cells out of 578 cells from no-shock control vmPFC (mouse #5, 140 out of 170; mouse #6, 125 out of 220; mouse #7, 57 out of 71; mouse #8, 55 out of 117), 384 cells out of 729 cells from ACC-vmPFC (mouse #9, 155 out of 325; mouse #10, 73 out of 108; mouse #11, 60 out of 148; mouse #12, 96 out of 148), 360 cells out of 688 cells from BLA-vmPFC (mouse #13, 95 out of 210; mouse #14, 58 out of 110; mouse #15, 93 out of 157; mouse #16, 114 out of 211). Further analyses were performed only on the selected cells. The peaks of calcium events were detected by applying a threshold (2.5 standard deviations of the calcium trace [df/f] from each cell), then the number of events was counted in 2-s bins (450 bins in total for 900-s long OF task) and used as the event rate. The calcium trace (df/f) of each cell was transformed into a z-score for further analysis.

### Identification of cluster-specific and component-specific neurons

We used a permutation test to identify the cluster-specific and component-specific neurons. We circularly shifted the timing of calcium events using a random number for each mouse 10,000 times and calculated the sum of the calcium events for each cluster and component. We calculated the p-value of each cell for each cluster or i1 and i2 + i3 components if the number of each cluster or component is more than 1% of all bouts, and after Holm-Bonferroni correction ($p < 0.05$, one-sided), named significant cells as cluster-specific or component-specific neurons.

### Decoding behavior from neural activity

We trained L2-regularized linear binary support vector machine (SVM) decoders to decode freezing behavior from neural activity patterns and one-versus-all multiclass error-correcting output code models to decode behavioral clusters using L2-regularized linear binary SVM decoders. The calcium event rate for each cell was partitioned into 2-s bins, and five consecutive bins corresponding to each 10-s bout from all imaged cells were used to train the SVM decoders. To decode freezing behavior, freezing rate were divided into two groups containing equal numbers of bouts by the median value to create binary labels. Tenfold cross-validation was implemented by randomly grouping 90 bouts into nine sets of nine bouts for training and one set of nine bouts for testing. Cross-validation was repeated 1000 times, and the decoding accuracy was calculated as the mean across repetitions. To evaluate the decoder accuracy, the decoder was trained using circularly shifted shuffled labels, and the average accuracy across 1000 shufflings was compared with the accuracy of the real data. A paired t-test was performed to compare the accuracy of real and shuffled decoders.

## Identification of shock-responding neurons

We calculated the mean df/f of the shock moment (0–2 s for each bout) and compared it to that of −2–0 s for each bout (60 bouts in total). If the mean df/f of 0–2 s of every bout was significantly bigger or smaller (paired $t$-test, $p < 0.05$) than that of −2–0 s, the cell was classified as shock-activated or shock-suppressed neurons.

## Identification of freezing correlated neurons

We calculated the Pearson's correlation between the freezing rate (2-s chunk) and each calcium event rate (2-s chunk) and set cells with a $p$-value of correlation <0.01 as significantly positively or negatively correlated neurons.

## Shock correlation

The shock correlation coefficient (Shock correlation) was calculated between the calcium event rate (2-s chunk) and 60 consecutive '1-0-0-0-0' arrays during the conditioning period. '1' here indicates the time chunk when the demonstrator receives foot shocks (shock moment), and '0' indicates the time chunks without shock.

## Time-series analysis of neural activity

To calculate the correlation of neural activity between adjacent bouts, the calcium event rate for each cell was partitioned into 2-s chunks, and Pearson's correlation coefficients were calculated between each bout. A paired $t$-test was performed to compare the activity correlation between bouts $n$ and $n + 1$ (2-s), $n + 2$ (4-s), …, to $n + 30$ (60-s).

## Spatial clustering analysis of annotated neurons

To evaluate the spatial distribution of annotated neurons (e.g., freezing positively correlated neurons), pairwise distances across all possible neuron pairs among the groups were calculated, and the mean value was compared with those from random shuffling to test its significance. Random shuffling was performed by selecting the same number of neurons as the original subpopulation, and the mean value of the pairwise distances was calculated, which was repeated 10,000 times. The real and the mean of the shuffled mean values were then compared using a paired $t$-test across mice.

## Statistics and reproducibility

No statistical methods were used to determine sample sizes. Suitable sample sizes were determined based on our previous experiences and similar studies which are generally employed in the field of study[7,24,63]. All the subject mice were randomly assigned to each experimental groups in each study. The nonlittermate demonstrator mice were randomly assigned to the observer mice. All the behavioral experiments were conducted with a blind group allocation during data collection and analysis, except for the in vivo Ca$^{2+}$ imaging experiments due to its high complexity. All animal behaviors were automatically tracked using DeepLabCut. All mice used for optogenetic inhibition and microendoscopy experiments were perfused and blindly post-hoc verified to include only individuals with appropriate expression and accurate optic fiber or GRIN lens implant position for further analysis. Only individuals who demonstrated less than 50% freezing during the habituation period (0–300 s) were included in the microendoscopy analysis. During microendoscopy, two mice had missing microendoscopy image frames (ACC-vmPFC mouse #10 [3 frames: 0.12 s], BLA-vmPFC mouse #14 [30 frames: 1.2 s]) at the end of the task due to technical problems. The last bouts from these individuals were omitted from the statistical analysis. To calculate shock correlation, a cell must show at least one calcium event during the conditioning period (301–900 s). Owing to the absence of calcium events during the conditioning period, four cells from the vmPFC group (Fig. 5g) and four cells from the BLA-vmPFC group (Fig. 6l) were excluded from the data analysis. All fluorescent image analyses were independently repeated at least twice and consistently demonstrated a similar trend.

## Reporting summary

Further information on research design is available in the Nature Portfolio Reporting Summary linked to this article.

## Data availability

The data generated in this study are provided in the Source Data file. Raw data will be provided upon request. Source data are provided with this paper.

## Code availability

The custom-made MATLAB script used for behavioral and calcium activity analysis can be downloaded from Github: https://github.com/okuyamalab/Huang_et_al_2023_NatCommun.

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

## Acknowledgements

We thank I. Yoshimura, M. Sako, A. Watarai, M.T. Tang, A. Matsuyama, and M. Watanabe for technical assistance, and all members of the Okuyama laboratory for discussion and support. We also thank Shinichiro Kira, Takahiro Ezaki, Dheeraj S. Roy, and Yasuko Isoe for research discussion. This work was supported by JST, PRESTO Grant Number JPMJPR1781 (to T.O.), JST FOREST Program, Grant Number JPMJFR2143 (to T.O.), JSPS KAKENHI Grant Numbers JP18H02544, JP20K21459, JP21H02593, and JP21H05140 (to T.O.), AMED under Grant Number JP21wm0525018 (to T.O.), Grant-in-Aid for JSPS postdoctoral PD fellowship JP20J01468 and JP19J00911 (to A.W. and M.W.), Grant-in-Aid for JSPS doctoral DC fellowship JP22J21085 and JP22J11822 (to Z.H. and M.C.), the Naito Foundation (to T.O.), and SECOM Science and Technology Foundation (to T.O.).

## Author contributions

Conceptualization, Z.H., M.C., and T.O.; Methodology, Z.H. and M.C.; Formal Analysis, Z.H., M.C., and K.T.; Investigation, Z.H., M.C., and K.T.; Writing – Original Draft, Z.H. and T.O.; Writing – Review & Editing, all coauthors; Resources, A.W., M.W., and H.I.; Supervision, T.O.

## Competing interests

The authors declare no competing interests.
