## [Peer review file · Nature Communications]

REVIEWER COMMENTS

Reviewer #1 (Remarks to the Author):

In this very interesting paper, the authors dissect the role of vmPFC and inputs to it from ACC and BLA, in responses by one mouse to the experience of footshock by a second mouse. The results reveal the existence of distinct subpopulations of neurons within the vmPFC that show activity correlated with different elements of observational fear: the animal's freezing versus escape, and the brief period when the conspecific was shocked. Among the paper's strengths are: 1) the use of unbiased DeepLabCut and machine learning methods to characterize behavior and corresponding neuronal ensemble activity, measured using calcium imaging; and 2) the combination of optogenetic inhibition of two inputs to the vmPFC and calcium imaging of the vmPFC to look at the function of these ensembles. This allows for causal inferences that could be made based on correlated neural activity alone.

The studies are systematic, the data are clear, and the findings are significant. The main weakness of the paper is the impenetrability of some portions of the text. As currently written, digesting the paper is heavy-going, even for someone familiar with the background.

Specific points:

1. The Discussion in particular is in need of rewriting to highlight the major findings and their relevance to the field in a clearer and more lucid fashion.
2. The ease of digesting the complex figures would in general be facilitated by more explicit figure captions. This reader found themselves frequently having to toggle back and forth between the results section and the figures in order to understand what was being presented. Clearer captions would make the figures more self-explanatory. For example: the caption of Figure 4a ("Bout-averaged responses of other-shock responding neurons") could be replaced with something like: "Change in activity of other-shock-responding neurons during 2-s shock bouts compared to preceding 2-s interval".
3. The workings of the model represented in 6J are not obvious and are not well explained either in the text or in the figure caption. Relatedly, on the basis of this figure, the Discussion refers to evidence of the existence of an "ACC-BLA network", whereas the data pertain only to the inputs of these of these structure to the vmPFC and not to any direct interaction.
4. The presentation of positive and negative correlations between neuronal activity of different subpopulations during self-freezing versus other-shock episodes seems overly complicated since some correlations can be inferred on the basis of others. For example, I'm not certain that the authors need to include Figures 4f and 4g, or at least not both of them, since the same story is essentially told – and more graphically – by Fig 4e.

5. Have the authors also studied female mice? Perhaps a statement should be added explaining the exclusive focus on males.

6. Some description of housing should be included in the methods. Presumably mice had to be housed singly after implantation of a headframe. But what about prior to that or in studies in which this second surgery was not performed? Were animals group- or singly housed, and if the former, were the paired mice in the experimental paradigm also cage-mates? This is important given prior work on differences between observational fear in mice reared together versus apart.

Reviewer #2 (Remarks to the Author):

In the submission "Ventromedial prefrontal neurons represent self-states shaped by vicarious fear" the authors address an interesting question, how vmPFC neurons represent behavioral responses induced by observing conspecific fear behavior. The authors employ state-of-the-art methodologies and experiments are for the most part nicely done. However, the lack of consistency in the analytic approach throughout the manuscript weakens the conclusions, some analysis and statements have to be revised to consider publication.

Please find below my comments for each part/figure.

1/As a first step, the authors aimed at characterizing and classifying the behavior of the observer mouse during an observational fear paradigm without taking into account the behavior of the demonstrator mouse. That seems a bit counterintuitive to me. Nevertheless, the choice of the paradigm and the method seems appropriate however the behavioral results are not conclusive. The analytic approach does not reveal stereotyped behavioral sequences but shows a mixture of distinct observer's posture, position and movement that for some tend to be more sustained during conditioning. Also, the approach does not allow to isolate and quantify the vicarious fear responses, i.e., freezing and escape behaviors. Furthermore, the behavioral classification described in figure 1 is poorly decoded by vmPFC activity and is not used by the authors to quantify the effects of optogenetic manipulations later. So, I am questioning the relevance of this analysis. At the end, the two main behavioral variables used by the authors are freezing and distance from the demonstrator side. These variables can be extracted by simply tracking the position of the mouse in the maze. The authors have to adjust the figure 1, either they go deeper with the analysis and they use it across the manuscript or they simplify the behavioral measurements and clarify their goal.

If the goal of the figure 1 is to reveal the full behavioral repertoire of the observer mouse then I don't understand why the author divided the behavioral session into 10 sec bouts corresponding to the foot shock frequency. It's the demonstrator mouse that receives the shock not the observer so why this

choice. As the authors mentioned: during these 10 sec time windows the demonstrator mouse behavioral sequence is apparently stereotyped (no quantification), composed of jumping and running around with vocalization followed by freezing. Furthermore, characterizing the behavior in 10 sec bouts is not accurate at all. If during these 10 sec bouts the behavior of the demonstrator mouse is variable why the behavior of the observer mouse would be stable? Is the observer's posture, position and movement homogeneous during a 10 sec bout? From figure 1f and 1g it does not seem to be the case so how the authors annotated each 10 sec behavior bouts describe in figure 1h is really not clear. If I consider a bout labeled as cluster 1 (red) does it mean that the mouse was only moving left during the 10 sec of this bout? I don't believe that's the case. Are the authors tried to embed the all session into 2D t-SNE space and cluster the data? Taking let say 13 body points x 2 coordinates x 900 sec of session x 2 frames/sec. How this data looks like? By this way the authors would obtain a more accurate behavioral characterization that is independent of the foot shocks intervals and would be able to extract stereotyped behavioral sequences at fine timescale.

In the same vein, after annotating each 10 sec bouts the authors examine the transition probability between them. The authors conclude that the patterns III, IV, V, VI, VII and VIII tend to be more maintained during conditioning than habituation. However, as described in figure 1f and quantify in figure 1g during these six patterns the mouse remained stationary mainly freezing. A simple comparison of the number of bouts for each cluster during habituation and conditioning would lead to the same results. It is surprising that the authors did not look at the transition probability between different clusters, by excluding intra cluster transitions. If you examine the matrix of the transition probability for conditioning on figure 1i, it seems that bout I (move left) can be followed bout V (left) or VII (near D, gazing). I would interpret that as a stereotyped behavioral sequence when the observer mouse is approaching the demonstrator side. Same for bouts II, VI and VIII when the observer mouse is moving away from the demonstrator side.

If the goal of the figure 1 is to isolate and quantify the vicarious fear responses it seems that the authors have the appropriate approach because they can measure freezing and label times when the mouse is near the demonstrator mouse (gazing) and far from it (escaped). They did it in figure 6. Could the authors quantify these three behaviors during habituation and conditioning for the observer mouse? And during the 10 sec of foot shock intervals? The camera is fast enough to have an accurate quantification of these behaviors. I will ask again this data for the figure 4 and 5. During conditioning, the author should see an increase in the time spend freezing (figure 1b) and escaped (control in figure 2h), and maybe a decrease of the gazing responses.

The legend of figure 1 d is not clear, is this t-SNE map contains data from one mouse or for all?

2/ The optogenetic inhibition of the vmPFC during conditioning reduces the time the observer mouse spends far from the demonstrator one (figure 2i). As I mentioned above, there is no need to classify behavior to reach that conclusion. It would be interesting to explore the transition probability between different clusters, by excluding intra cluster transitions.

3/ The authors performed calcium imaging of vmPFC neurons to reveal the neural representation of the observer mouse behavior during the OF task. Here again, the methodology is appropriate and the experiments well done but the analysis and results are weak and most of the conclusions are close to the overstatement.

The 3 results are: 1) 2% of neurons are activated in individual clusters. 2) using decoders to distinguish among the eight types of clusters based on the pattern of vmPFC population activities the best decoding performance reach is below 50%, so more than 50% of the time the decoder is wrong. 3) freezing can be decoded using vmPFC population activities, that is already known. The authors' interpretation of these results is: "Taken together, these results suggest that the vmPFC population neurophysiologically represents multifaceted behavioral states of the self, including both escape and freezing behavior, which were specifically induced by OF." This is clearly an overstatement. Maybe by refining the analysis, specially the behavioral one, the authors could reach such conclusion.

Regarding the first analysis, if I understand it well, the authors identified neurons activated during individual cluster. Why is this analysis so exclusive? It would have been surprising that prefrontal neurons can discriminate between behaviors distinguishing body orientation like III, IV, V or VI. Maybe head-direction neurons could do it. Furthermore, the behaviors III, IV, V or VI have nothing to do with behavioral states of the self.

The authors should focus on escape and freezing behavior. The authors did it for freezing but not for escape behavior. I think that the main reason is because of the analytic approach to detect escape behavior. With an accurate detection of escape behavior, the authors will be able to identify its associated neuronal dynamics. Are prefrontal neurons activated at the initiation, the execution or the end of the escapes? Also knowing the role of prefrontal in social behavior, are neurons activated at the gazing instances? Can the authors compare the activity evoke by each individual cluster during habituation and conditioning? The same comments apply for the population analysis.

4/ In this part the authors identified shock-activated and -inhibited neurons. These neurons show transient increase or decrease of activity at the time of the shock when the demonstrator mouse is apparently (no quantification) jumping and running around with vocalization followed by freezing. Unfortunately, we ignore what is the behavior of the observer mouse during the same time window: 2sec before and 10 sec after shock? I would like to see the freezing, the speed of a body point let say the back or nose and the body orientation in comparison to the observer side during this time window. If the behavioral sequence is similar to the demonstrator mice escape followed by freezing it would be good to re-interpret the results of the figure. Furthermore, to ensure a continuity of the analysis across the manuscript it would be interesting to compare the responses of other-shock responding neurons to the 8 different types of behavioral bouts. Are shock-activated neurons activated during bout II when the mouse is moving to the right? For example.

As a second analysis, the authors identified freezing positively- and negatively-correlated neurons. Interestingly, freezing positively-correlated neurons are inhibited during shock and freezing negatively-

correlated neurons are activated, like the shock-inhibited and -activated neurons. Are freezing positively-correlated neurons the same neurons than shock-inhibited ones? And, are freezing negatively-correlated neurons the same neurons than shock-activated ones? I could not find the overlapping values in the manuscript. If the overlap is high, the authors have to re-interpret the results of the figure.

In figure 4a and 4e what is the black line refer to?

5/ This is a really interesting sets of experiments. However, I think that the data is poorly exploited. The authors can do better. If the proposed manipulations affect the link between other-shock and self-freezing activities then the decoding performance of behavioral clusters and freezing behavior should be affected. The authors should do this analysis here.

6/ This is a really interesting sets of experiments. However, the authors did not use the same analytic approach described in figure 1 and they quantified escape behavior with a different strategy than in figure 2i. This lack of consistency in the methods across the manuscript do not allow the reader to trust the results and the proposed model in figure 6j.

Reviewer #3 (Remarks to the Author):

In this manuscript the authors investigate the neural representation in the vmPFC of a mouse self-state related to a conspecific receiving repetitive foot shocks, using an observational fear paradigm. In this task, they made an interesting classification of stereotypic behavioural pattern thus increasing the number of observations usually limited, in the current literature, to measurement of freezing behavior. Building on previous reports showing the involvement of brain regions such as the ACC and the BLA in the acquisition of vicarious fear, the authors made an optogenetic study extending the role of these regions in a circuit that involves the vmPFC. Specifically the authors show:

They made a noteworthy behavioral classification, presented in Figure 1, showing that this approach can detect other behaviors than freezing, such as escape (closet VIII). However, in Figure 6, to show the role of ACC-vmPFC and BLA-vmPFC circuits in freezing vs. escape behavior, the authors used a different behavioral analyses (far vs. near side of the cage). Thus is not clear why the authors elaborated such sophisticated behavioural analysis.

Also related to the behavioral analysis, it not clear how 'back' (cluster III) is different from 'escape' (cluster VIII). A mouse avoiding gazing his conspecific could also be considered an escape behavior. Indeed, in figure 2 it seems that optogenetic inhibition of the vmPFC also reduced cluster III.

The identified behavioral clusters are then used to study the neurophysiology related to these events. This is very interesting and the authors showed that vmPFC activity is also specific for the behavioral cluster. However, figure 3 showed that only 7 cells were activated by individual clusters. The authors used n=4 mice for this analysis, thus, this means that even less cells were activated for each animal? and in some animals no cells were activated for the clusters? Thus it is difficult to assess how relevant is the activation of these cells for the subsequent measurements in the manuscript.

In the following paragraph the authors described shock-activated and -suppressed cells and freezing-positive and -negative cells in the vmPFC and in the circuits including the ACC and the BLA. It is not clear, how the authors reconcile the observation of self-freezing cells in the vmPFC with the fact that perturbation of this area do not affect freezing behavior.

Moreover, after the initial description of avoidance behavior, the authors then jump to other-shock and self-freezing encoding in the vmPFC and is not clear how this is linked to the escape behavior and to the activity of the vmPFC cells during the behavioral clusters.

Other comments related to the manuscript:

- Observers' freezing behavior during habituation seems significantly increased compared to the demonstrator? are the demonstrators familiar? do the observers received previous exposure to the shocks?
- Why do the authors used different shock intensities across experiments?
- In Fig. 2, for the optogenetic inhibition the authors applied 13 mW green light during the OF task, while 5mW for in vivo single unit recording. Why do the authors used different light intensities?

Point-by-point responses:

Reviewer #1:

In this very interesting paper, the authors dissect the role of vmPFC and inputs to it from ACC and BLA, in responses by one mouse to the experience of footshock by a second mouse. The results reveal the existence of distinct subpopulations of neurons within the vmPFC that show activity correlated with different elements of observational fear: the animal's freezing versus escape, and the brief period when the conspecific was shocked. Among the paper's strengths are: 1) the use of unbiased DeepLabCut and machine learning methods to characterize behavior and corresponding neuronal ensemble activity, measured using calcium imaging; and 2) the combination of optogenetic inhibition of two inputs to the vmPFC and calcium imaging of the vmPFC to look at the function of these ensembles. This allows for causal inferences that could be made based on correlated neural activity alone.

The studies are systematic, the data are clear, and the findings are significant. The main weakness of the paper is the impenetrability of some portions of the text. As currently written, digesting the paper is heavy-going, even for someone familiar with the background.

Specific points:

1. The Discussion in particular is in need of rewriting to highlight the major findings and their relevance to the field in a clearer and more lucid fashion.

Response:

We thank the reviewer for the careful assessment of the manuscript and the positive comments acknowledging the strength of the data and the value of this study for the field. In the **Discussion** section, we added numbering and explanation in the first paragraph to highlight our two main findings. In addition, we added discussions based on our new 2-s behavior component analysis we performed according to reviewer 2 & 3's comments.

(p.12, line 288-290)

"Here, we primarily discuss the (1) neural mechanisms underlying OF-induced escape behavior and (2) vmPFC neural representation in the OF on self-states and other-states."

(p.12, line 296-299)

"... a t-SNE-based clustering analysis using body-point data tracked by DeepLabCut automatically classified eight types of stereotypic 10-s behavior clusters and 2-s behavior components during the OF task that could not be segmented by the freezing rate alone (Fig. 1h, 2d). Eight behavior clusters consisted of different characteristic behavior components (Fig. 2f)."

(p.13, line 332-340)

"Although the decoding accuracy of behavioral clusters was higher than that of the random data, the overall rate remained below 50 %. This could be because the 10-s bouts

contained a mix of components, albeit with varying proportions (Fig. 2f). However, our analysis did reveal the presence of neurons that were specifically active during the escaped (i1) and gazing (i2+i3) behaviors (Fig. 4j), which are major components of freezing positively-correlated neurons (Supplementary Fig. 10b). We thus speculate that these distinct components underlie the significantly higher decoding rate for eight clusters compared to the shuffled data. Although it would be unreliable to decode those behavior components with the unbalanced dataset in its current form, such analysis may provide more evidence to support this hypothesis with a larger dataset.”

2. The ease of digesting the complex figures would in general be facilitated by more explicit figure captions. This reader found themselves frequently having to toggle back and forth between the results section and the figures in order to understand what was being presented. Clearer captions would make the figures more self-explanatory. For example: the caption of Figure 4a (“Bout-averaged responses of other-shock responding neurons”) could be replaced with something like: “Change in activity of other-shock-responding neurons during 2-s shock bouts compared to preceding 2-s interval”.

Response:

Thank you for the suggestion. We changed the figure legend as suggested, and also added the black line information as reviewer 2 asked about it.

Fig.5 legend (p.31, line 899-900, 902-904), Fig.6 legend (p.31, line 916-920, 927, p.32, line 928-931)

“Change in activity of other-shock responding neurons during 2-s shock bouts compared to preceding 2-s interval. The black line represents the n.s. group.”

3. The workings of the model represented in 6J are not obvious and are not well explained either in the text or in the figure caption. Relatedly, on the basis of this figure, the Discussion refers to evidence of the existence of an “ACC-BLA network”, whereas the data pertain only to the inputs of these of these structure to the vmPFC and not to any direct interaction.

Response:

The model presented in the previous Fig. 6J (new **Fig. 7I**) serves as a summarizing figure of our findings, and we addressed this model in the **Discussion** section (p.14, line 364-367). We agree that the previous illustration of an “ACC-BLA network” was misleading since we did not provide any data for this network. Therefore, we changed the black line between ACC and BLA into a gray dashed line to make it clear that this part is suggested in previous studies.

4. The presentation of positive and negative correlations between neuronal activity of different subpopulations during self-freezing versus other-shock episodes seems overly complicated since some correlations can be inferred on the basis of others. For example, I’m not certain that the authors need to include Figures 4f and 4g, or at least not both of them, since the same story is essentially told – and more graphically – by Fig 4e.

Response:

We agree that there is a strong connection between the information on neuronal activity and correlation, and one can be inferred from the other. However, we believe it is important to examine the overall correlation of identified neurons (i.e. self-freezing correlated neurons and other-shock responding neurons) to confirm the population characteristics since the overlap between these neurons was not complete (**Supplementary Fig. 9j**). At the population level, these figures demonstrate the mixed representation of the two types of information.

5. Have the authors also studied female mice? Perhaps a statement should be added explaining the exclusive focus on males.

Response:

We did not use female mice in this study. In our study, we attempted to eliminate factors that affect the variance of animals' behavior. It is known that female mice's social behaviors can be altered depending on the estrous cycle (Palanza et al., *Physiol Behav*, 2001; Chen et al., *Behav Brain Res*, 2009). In addition, a study reported lower freezing behavior of both demonstrator and observer female rats in the OF task (Han et al., *Sci Rep*, 2020). Considering these reports, we fixed the experimental conditions with young adult male mice here.

We have provided this information about sex in the **Methods** section (p.15, line 384-385) and added it to the **Abstract** section (p.2, line 21-22).

"Here, our automatic behavior classification detected the male observer's stereotypic behaviors during observational fear (OF)."

6. Some description of housing should be included in the methods. Presumably mice had to be housed singly after implantation of a headframe. But what about prior to that or in studies in which this second surgery was not performed? Were animals group- or singly housed, and if the former, were the paired mice in the experimental paradigm also cage-mates? This is important given prior work on differences between observational fear in mice reared together versus apart.

Response:

The subject (observer) mice were group-housed both before and after the surgery. The mice used for optogenetic manipulation and Ca²⁺ imaging underwent a single surgery which involved virus injection and optic fiber or GRIN lens implantation. After the surgery, the subject mice were returned to their home cage and socially housed with their cage mates until seven days prior to the behavioral experiment. The subject mice were then co-housed with a novel mouse that would later serve as the demonstrator mouse. We described the details about housing and other information in the **Methods** section (p.17, line 463-464).

"The mice that underwent optic fiber, or GRIN lens implantation surgery, were co-housed immediately after the surgery."

Reviewer #2:

In the submission "Ventromedial prefrontal neurons represent self-states shaped by vicarious fear" the authors address an interesting question, how vmPFC neurons represent behavioral responses induced by observing conspecific fear behavior. The authors employ state-of-the-art methodologies and experiments are for the most part nicely done. However, the lack of consistency in the analytic approach throughout the manuscript weakens the conclusions, some analysis and statements have to be revised to consider publication.

Please find below my comments for each part/figure.

1/As a first step, the authors aimed at characterizing and classifying the behavior of the observer mouse during an observational fear paradigm without taking into account the behavior of the demonstrator mouse. That seems a bit counterintuitive to me. Nevertheless, the choice of the paradigm and the method seems appropriate however the behavioral results are not conclusive. The analytic approach does not reveal stereotyped behavioral sequences but shows a mixture of distinct observer's posture, position and movement that for some tend to be more sustained during conditioning. Also, the approach does not allow to isolate and quantify the vicarious fear responses, i.e., freezing and escape behaviors.

Response:

We appreciate the positive evaluation of the experimental design and constructive comments on the manuscript. We have performed additional analysis to address the concerns raised by the reviewer and we feel that our manuscript has been substantially improved through this revision. As the reviewer pointed out, we analyzed the observer's behavior without taking into account the demonstrator's behavior. However, we performed the behavior analysis in 10s length according to the periodic behavioral reactions in every 10s of the demonstrators.

We agree that our behavioral sequence analysis, which used 10-s bouts, was a mixture of poses, positions, and movements. In order to focus on more detailed vicarious fear responses and explore behavior changes due to optogenetic manipulations and neural representations, we performed a new analysis aimed at extracting vicarious fear responses with shorter time windows (2-s, **Fig. 2, Supplementary Fig. 4**). We successfully isolated one "escaped" and two "gazing"-related behavior components in the new analysis. As suggested by the reviewer, we applied this analysis throughout the manuscript to ensure consistency.

In the **Result** section (p.6, line 112-124), we added the following paragraph corresponding to Figure 2.

"We conducted a similar analysis using shorter 2-s bouts to isolate the vicarious fear responses further. As mice move and stop continuously, we set a threshold of 50 % in the 2-s freezing rate to classify bouts as mobile or immobile. We then performed unsupervised classification and obtained 9 components for each dataset (Fig. 2a,b, Supplementary Fig. 4a,b). Among all the components (Supplementary Fig. 4c,d), three characteristic

components were extracted from the mobile dataset, namely (m1) approaching, (m2) leaving, (m3) moving near D, three from the immobile dataset, (i1) escaped, (i2) gazing-1, (i3) gazing-2 (Fig. 2c). The freezing rate of m1 and m2 were significantly lower compared to other mobile components (Fig. 2d, Supplementary Table). When the habituation and conditioning period were compared, the proportion of mobile components (m1–m3) decreased, while the immobile components (i1–i3) increased (Fig. 2e). Among the eight clusters obtained in Fig. 1, m1 shared the largest portion of cluster I, m2 of cluster II, m3 of cluster I and VII, i1 of cluster III and VIII, i2 of V, and i3 of VII (Fig. 2f) indicating that the 10-s clusters contained a mixture of different behavior components.”

Furthermore, the behavioral classification described in figure 1 is poorly decoded by vmPFC activity and is not used by the authors to quantify the effects of optogenetic manipulations later. So, I am questioning the relevance of this analysis.

Response:

While the vmPFC decoder using the real calcium data performed significantly better than the shuffled data, it is true that the decoder accuracy was not high (below 50 %). We will address this point in detail in the Fig. 3 section. Additionally, we applied the new 2-s analysis to all the data we present in the current study to ensure consistent evaluation, and we will discuss it in detail in each section.

At the end, the two main behavioral variables used by the authors are freezing and distance from the demonstrator side. These variables can be extracted by simply tracking the position of the mouse in the maze. The authors have to adjust the figure 1, either they go deeper with the analysis and they use it across the manuscript or they simplify the behavioral measurements and clarify their goal.

Response:

We consider our new analysis based on 2-s bouts to provide a more accurate measurement of vicarious fear than solely tracking the position of the mouse, as it offers better temporal resolution in terms of behavior components (**Supplementary Fig.8**). We will discuss this point further in the Fig.3 section. Furthermore, by applying the same analysis to the optogenetics part (new **Fig. 3 & 7**), we were able to strengthen our previous conclusion in a more refined and consistent manner. Overall, the new analysis has enriched our behavior analysis, moving beyond solely using freezing and distances, and providing more focused and detailed descriptions of escape and gazing behaviors.

If the goal of the figure 1 is to reveal the full behavioral repertoire of the observer mouse then I don't understand why the author divided the behavioral session into 10 sec bouts corresponding to the foot shock frequency. It's the demonstrator mouse that receives the shock not the observer so why this choice. As the authors mentioned: during these 10 sec time windows the demonstrator mouse behavioral sequence is apparently stereotyped (no quantification), composed of jumping and running around with vocalization followed by freezing.

Response:

Although the observer's behavior is distinguishable both in 10-s cluster and 2-s components, the cue given to the demonstrator, which could alter the observer's behavior, is provided every 10 s. This is why we wanted to investigate how the 10-s bout changes in observer mice with increasing cues provided by the demonstrator over time.

Furthermore, characterizing the behavior in 10 sec bouts is not accurate at all. If during these 10 sec bouts the behavior of the demonstrator mouse is variable why the behavior of the observer mouse would be stable? Is the observer's posture, position and movement homogeneous during a 10 sec bout? From figure 1f and 1g it does not seem to be the case so how the authors annotated each 10 sec behavior bouts describe in figure 1h is really not clear. If I consider a bout labeled as cluster 1 (red) does it mean that the mouse was only moving left during the 10 sec of this bout? I don't believe that's the case.

Are the authors tried to embed the all session into 2D t-SNE space and cluster the data? Taking let say 13 body points x 2 coordinates x 900 sec of session x 2 frames/sec. How this data looks like? By this way the authors would obtain a more accurate behavioral characterization that is independent of the foot shocks intervals and would be able to extract stereotyped behavioral sequences at fine timescale.

Response:

We apologize for the confusion. To make the analysis process clearer, we have added a schematic drawing explaining the 10s analysis in **Fig. 1d** and 2s analysis in **Fig. 2a**. For the 10 s analysis, we obtained 13 mice x 90 bouts (1 bout = 10 s) = 1,170 bouts, which are 1,950 D (10 sec x 7.5 frames/s x 13 body points x 2 coordinates (x,y)). Then, by performing t-SNE embedding, we obtained 1,170 bouts that are 2D. This embedding result is shown in **Fig. 1e,f**, where each bout is indicated as one point. Therefore we obtained cluster labels for each bout. In **Fig. 1i**, using the obtained 1,170 cluster labels, we drew the behavior cluster sequence of each mouse. Also, we agree that each 10-s bout cluster contains heterogeneous behavior components. Therefore, we performed a 2-s bout analysis and succeeded in extracting stereotyped behavior components more accurately, such as m1 "approaching", m2 "leaving", i1 "escaped", and i2 & i3 "gazing" (**Fig. 2c**). We confirmed that the clusters we obtained contained mixed components, but there was consistency between the clusters and the components. For example, m1 "approaching" was the main component of the cluster I "moving left" and i1 "escaped" was the main component of the cluster III "far from D" (**Fig. 2f**).

However, as there were other behavior components in clusters VII and VIII besides gazing and escaped behaviors, we removed the names from clusters VII and VIII (**Result**, p.5 line 91-93).

"... we annotated each behavioral cluster as follows: (I) move left, (II) move right, (III) back, (IV) front, (V) left, (VI) right, (VII) near a demonstrator (D) (~~gazing~~), and (VIII) far from a demonstrator (D) (~~escaped~~)"

In the same vein, after annotating each 10 sec bouts the authors examine the transition probability between them. The authors conclude that the patterns III, IV, V, VI, VII and VIII tend to be more maintained during conditioning than habituation. However, as described in figure 1f and quantify in figure 1g during these six patterns the mouse remained stationary mainly freezing. A simple comparison of the number of bouts for each cluster during habituation and conditioning would lead to the same results. It is surprising that the authors did not look at the transition probability between different clusters, by excluding intra cluster transitions. If you examine the matrix of the transition probability for conditioning on figure 1i, it seems that bout I (move left) can be followed bout V (left) or VII (near D, gazing). I would interpret that as a stereotyped behavioral sequence when the observer mouse is approaching the demonstrator side. Same for bouts II, VI and VIII when the observer mouse is moving away from the demonstrator side.

Response:

We performed the transition probability analysis in **Supplementary Fig. 2b** (previously Fig. 1i), but could not obtain a stable result for between-cluster transitions. This is reasonable, as pointed out by the reviewer, each cluster consists of different 2-s components (**Fig. 2f**). Therefore, we decided not to perform further transition analysis by excluding intra-cluster transitions. What we still can tell here is that, although we did not find common transitions in all individuals, during the conditioning period, once a behavior is transitioned into one of the clusters with a high percentage of freezing (i.e. clusters III to VIII), it does not change from there (i.e., more repetition). Thus, the transitions mentioned by the reviewer may exist, but we could not confirm them through cluster transition analysis. Moreover, although we attempted to perform this transition analysis using 2-s components, the results were unreliable since the overall number of bouts was not large enough for the $18 \times 18 = 324$ transition patterns, and the number of transitions varied significantly among transition patterns, which could overweight the rare events when tested statistically.

If the goal of the figure 1 is to isolate and quantify the vicarious fear responses it seems that the authors have the appropriate approach because they can measure freezing and label times when the mouse is near the demonstrator mouse (gazing) and far from it (escaped). They did it in figure 6. Could the authors quantify these three behaviors during habituation and conditioning for the observer mouse? And during the 10 sec of foot shock intervals? The camera is fast enough to have an accurate quantification of these behaviors. I will ask again this data for the figure 4 and 5. During conditioning, the author should see an increase in the time spend freezing (figure 1b) and escaped (control in figure 2h), and maybe a decrease of the gazing responses.

Response:

As the reviewer pointed out, our goal for figure 1 is to isolate and quantify vicarious fear responses. This was only partially achieved with the 10-s bout analysis, but our new 2s analysis allowed us to successfully extract escaped (i.e., i1) and gazing (i.e., i2 & i3) behavior with fine time resolution. We have applied this new analysis to the remaining parts, and we will discuss them in later sections one by one. Through detailed analysis, we observed that the proportion of immobile components increased in the conditioning

period, including i1 “escaped”, i2 “gazing1” and i3 “gazing2”, while the proportion of big movements (ie., m1 “approaching” and m2 “leaving”) decreased (**Fig. 2e**).

The legend of figure 1 d is not clear, is this t-SNE map contains data from one mouse or for all?

Response:

We would like to apologize for any confusion that may have arisen from the figure. Please note that this map contains data from all mice. To enhance the figure's clarity, we have included an illustration that explains the analysis shown in **Fig. 1d**, and we have also refined the figure legends.

Fig. 1 legend (p.29, line 836) “... (data from $n = 13$ mice, 1,170 bouts total) ”

2/ The optogenetic inhibition of the vmPFC during conditioning reduces the time the observer mouse spends far from the demonstrator one (figure 2i). As I mentioned above, there is no need to classify behavior to reach that conclusion. It would be interesting to explore the transition probability between different clusters, by excluding intra cluster transitions.

Response:

Using the 2-s analysis, we found that the i1 “escaped” component decreased, and the “gazing” components i2+i3 increased significantly in the inhibition group compared to the control group (**Fig. 3h,i**). While these findings are in line with the simple comparison of position in the x-coordinate (**Fig. 3j**), the new 2-s analysis provides richer information on the exact behavior component, which reflects not only different positions but also different postures or behavioral meanings. We didn’t include the transition analysis for the reasons we mentioned above.

We changed the following sentence in the **Result** section (p.6 line 139-141).

“In line with this, the proportion of the i1 “escaped” was significantly smaller and i2+i3 “gazing” was significantly larger in the inhibition group compared to the control group (Fig. 3h–j). To further confirm this trend, ...”

3/ The authors performed calcium imaging of vmPFC neurons to reveal the neural representation of the observer mouse behavior during the OF task. Here again, the methodology is appropriate and the experiments well done but the analysis and results are weak and most of the conclusions are close to the overstatement.

The 3 results are: 1) 2% of neurons are activated in individual clusters. 2) using decoders to distinguish among the eight types of clusters based on the pattern of vmPFC population activities the best decoding performance reach is bellow 50%, so more than 50% of the time the decoder is wrong. 3) freezing can be decoded using vmPFC population activities, that is already known. The authors’ interpretation of these results is: “Taken together, these results suggest that the vmPFC population neurophysiologically represents multifaceted behavioral states of the self, including both escape and freezing behavior, which were specifically induced by OF.” This is

clearly an overstatement. Maybe by refining the analysis, specially the behavioral one, the authors could reach such conclusion.

Response:

Thank you for the thoughtful suggestions and comments on this section. We will address point 1) cluster neurons in the next comment. Regarding point 2) the vmPFC decoding performance, we agree with the reviewer that while the decoding performance was statistically above the shuffled data, the overall decoding performance is not good enough, as it is below 50%. As the reviewer pointed out, and also as we showed, there are several unseparated behavioral segments in the 10-s bouts (**Fig. 2f**). The insufficient time resolution for accurate behavior classification could result in low performance. However, since each cluster was mixed yet contained specific behavior components, we believe this is why the cluster information is partially able to be decoded. We could not go deeper with the decoding of all 2-s components to test this hypothesis because the result is not very likely to be reliable due to the unbalanced data distribution of the components.

We added these discussions in the **Discussion** section (p.13, line 332-340).

“Although the decoding accuracy of behavioral clusters was higher than that of the random data, the overall rate remained below 50 %. This could be because the 10-s bouts contained a mix of components, albeit with varying proportions (Fig. 2f). However, our analysis did reveal the presence of neurons that were specifically active during the escaped (i1) and gazing (i2+i3) behaviors (Fig. 4j), which are major components of freezing positively-correlated neurons (Supplementary Fig. 10b). We thus speculate that these distinct components underlie the significantly higher decoding rate for eight clusters compared to the shuffled data. Although it would be unreliable to decode those behavior components with the unbalanced dataset in its current form, such analysis may provide more evidence to support this hypothesis with a larger dataset. ”

Also, we agree that it was an overstatement with our previous data to argue “Taken together, these results suggest that the vmPFC population neurophysiologically represents multifaceted behavioral states of the self, including both escape and freezing behavior, which were specifically induced by OF.” However, our new data on the presence of i1 escaped neurons (and i2+i3 gazing neurons) shown in **Fig. 4j** supports this sentence. We will discuss those component-specific neurons in the next section in detail. Thus, we have decided to keep this sentence in the revised manuscript.

Regarding the first analysis, if I understand it well, the authors identified neurons activated during individual cluster. Why is this analysis so exclusive? It would have been surprising that prefrontal neurons can discriminate between behaviors distinguishing body orientation like III, IV, V or VI. Maybe head-direction neurons could do it. Furthermore, the behaviors III, IV, V or VI have nothing to do with behavioral states of the self.

Response:

Thank you for the detailed discussion on cluster-specific neurons. We agree that our previous analysis was too exclusive with the Bonferroni correction. Also, we reconsidered the criteria of $\alpha = 0.025$ (two-sided), as it is difficult to discuss the cluster-negative neurons with the number of firing using the calcium imaging data. For these two reasons, we renewed our analysis using $\alpha = 0.05$ with Holm-Bonferroni correction (changed in the **Methods** section as shown below). We identified 11.5% of cells as cluster-specific neurons in this way (**Fig. 4e**). While there was no overlap between III, IV, V, and VI neurons, one neuron responded to both V and VIII. This overlap may reflect shared behavior components other than m1-m3 and i1-i3 (**Fig. 2f**). As our video was recorded from one direction and could not accurately calculate the direction, it is difficult to investigate whether there are head-direction neurons or not. However, we were at least able to identify i1 and i2+i3 specific neurons (also, two neurons were overlapped, **Fig. 4j**). While clusters III, IV, V, and VI seemed unrelated to the OF-specific behavior, i1, i2, and i3 behavior component were included in different proportions in those four clusters (**Fig. 2f**). Therefore, it is possible that activity of those component-specific neurons underlies the activity of cluster III, IV, V, and VI neurons.

We made changes to the **Result** section regarding cluster-specific neurons as follows (p.7, line 156-158).

“Consequently, we found that 11.5 % of cells (41 of 355) ~~only a few cells (7 of 355, 2.0%)~~ were significantly activated in individual clusters ~~(none in clusters V and VIII)~~ (Fig. 4e), while 9.5 % of cells (36 of 377) were identified as cluster-specific neurons in the no-shock control group (Supplementary Fig. 6d,e).”

We have added the following sentences to the **Result** section regarding i1 and i2+i3 specific neurons in (p.7, line 168-173).

“We further investigated the 2-s behavior component information encoding in the vmPFC. Since two components, i1 and i2+i3, showed significant decrease and increase, respectively, in the vmPFC inhibition group compared to the control group (Fig. 3), we hypothesized these information is particularly represented in vmPFC neurons. We found 17 of 355 neurons significantly active at the component i1 and 11 of 355 at the component i2+i3 with 2 overlapping neurons (Fig. 4j).”

We also added the following sentences to the **Methods** section (p.21, line 603-606).

“We calculated the p-value of each cell for each cluster or i1 and i2+i3 components if the number of each cluster or component is more than 1 % of all bouts, and after Holm-Bonferroni correction ($p < 0.05$, one-sided), named significant cells as cluster-specific or component-specific neurons.”

The authors should focus on escape and freezing behavior. The authors did it for freezing but not for escape behavior. I think that the main reason is because of the analytic approach to detect escape behavior. With an accurate detection of escape behavior, the authors will be able to identify its associated neuronal dynamics. Are prefrontal neurons activated at the initiation, the execution or the end of the escapes? Also knowing the role of prefrontal in social behavior, are

neurons activated at the gazing instances? Can the authors compare the activity evoked by each individual cluster during habituation and conditioning? The same comments apply for the population analysis.

Response:

With our new 2-s analysis, we were able to detect neurons specifically active at the timing of escaped behavior (component i1). By aligning the calcium activity by the onset and offset, we confirmed that the i1-specific neurons were activated at the timing of execution (**Supplementary Fig. 10k**). For i2+i3 gazing neurons, we found a similar tendency (**Supplementary Fig. 10l**).

In the **Result** section (p.9, line 219-227), we have added the following sentences about the neural properties of i1 and i2+i3 specific neurons.

“Furthermore, we investigated the neural properties of i1 “escaped”- specific and i2+i3 “gazing”-specific neurons, ... Notably, we confirmed that the activity of the i1-specific neurons was high at the timing of component i1 compared to the preceding and proceeding 2-s bouts and component i2+i3 compared to the proceeding 2-s bouts (Supplementary Fig. 10k,l).”

4/ In this part the authors identified shock-activated and -inhibited neurons. These neurons show transient increase or decrease of activity at the time of the shock when the demonstrator mouse is apparently (no quantification) jumping and running around with vocalization followed by freezing. Unfortunately, we ignore what is the behavior of the observer mouse during the same time window: 2sec before and 10 sec after shock? I would like to see the freezing, the speed of a body point let say the back or nose and the body orientation in comparison to the observer side during this time window. If the behavioral sequence is similar to the demonstrator mice escape followed by freezing it would be good to re-interpret the results of the figure.

Response:

We thank the reviewer for this important comment regarding the behavior of the observer. To clarify this point, as suggested, we compared the freezing and speed data to the x-coordinate of the observer every 2 s (**Supplementary Fig. 8a,b**). While the freezing was significantly reduced during the shock, the average speed remained stable over time. Although we were unable to accurately measure the real moving speed and orientation due to our recording limitations (i.e., we only captured data from one direction and could not record all body points), our 2-s behavior component analysis revealed that the proportion of immobile components was lowest during the 0-2 s, while the observer's moving components (i.e., m1 “approaching” and m2 “leaving”) were detected during the interval (**Supplementary Fig. 8c**). Therefore, it is not always escape followed by freezing. Rather, it seems that the observer changes position dynamically by approaching and leaving in response to the demonstrator's cue at the shock timing over the 10-min conditioning period.

We have added the following sentences to the **Result** section (p.8, line 188-193)

“We observed periodic features in the observer’s behavioral changes during each shock frequency (10 s) (Supplementary Fig. 8). Although freezing rate was significantly lower at the shock timing (0–2 s, Supplementary Fig. 8a), the average movement speed along the x-axis had no temporal feature (Supplementary Fig. 8b). These behavioral features indicate that the observer’s behavior is dependent on demonstrator being shocked. Thus, to ...”

We also added the following sentences to the **Methods** section (p.18, line 508-510)

“The speed of each mouse during the OF task was calculated by summing the differences in the x coordinates of the back center of each frame during a 2-s bout.”

Furthermore, to ensure a continuity of the analysis across the manuscript it would be interesting to compare the responses of other-shock responding neurons to the 8 different types of behavioral bouts. Are shock-activated neurons activated during bout II when the mouse is moving to the right? For example.

Response:

We would like to thank the reviewer for the valuable suggestions. We compared the responses of other-shock responding neurons to the eight different clusters (**Response Figure 1a**). Although we observed shock-activated neurons activated during cluster II, shock-activated neurons also activated in other clusters. We also compared the responses of self-freezing correlated neurons to the eight different clusters (**Response Figure 1b**), but we were unable to conclude whether the annotated neurons activated or suppressed more in any clusters.

Response Figure 1

a, Change in activity of other-shock responding neurons during each cluster (Cluster I – VIII). Green line, shock-activated neurons; yellow line, shock-suppressed neurons; black line, n.s. neurons. **b**, Change in activity of self-freezing correlated neurons during each cluster (Cluster I – VIII). Red line, freezing-positively correlated neurons; blue line, freezing-negatively correlated neurons; black line, n.s. neurons. Data are presented as mean \pm SEM.

As a second analysis, the authors identified freezing positively- and negatively-correlated neurons. Interestingly, freezing positively-correlated neurons are inhibited during shock and freezing negatively-correlated neurons are activated, like the shock-inhibited and -activated neurons. Are freezing positively-correlated neurons the same neurons than shock-inhibited ones? And, are freezing negatively -correlated neurons the same neurons than shock-activated ones? I could not find the overlapping values in the manuscript. If the overlap is high, the authors have to re-interpret the results of the figure.

Response:

Thank you for the comment. We added a matrix that depicts the overlapping populations of freezing-correlated and shock-responding neurons (**Supplementary Fig. 9j**). We found that approximately half of the shock-activated neurons were freezing negatively correlated neurons, and the shock-negatively correlated neurons were freezing positively correlated. However, there is no complete overlap between the two, which suggests that the information conveyed by each type of neuron is distinct, which was also suggested by the behavior (**Supplementary Fig. 8**) as we discussed above.

We added this part in the **Result** section (p.9, line 213-215).

“While there were some overlaps between shock-responding neurons and freezing-correlated neurons, not all neurons exhibited both representations (Supplementary Fig. 9j).”

In figure 4a and 4e what is the black line refer to?

Response:

We apologize for the lack of explanations for this part. The black lines represent cells that do not belong to either group (i.e., n.s. cell group).

Together with the comment from reviewer 1, we changed this explanation in the Fig.5 legend Fig.5 legend (p.31, line 899-900, 902-904) and Fig.6 legend (p.31, line 916-920, 927, p.32, line 928-931).

“Change in activity of other-shock responding neurons during 2-s shock bouts compared to preceding 2-s interval. The black line represents the n.s. group.”

5/ This is a really interesting sets of experiments. However, I think that the data is poorly exploited. The authors can do better. If the proposed manipulations affect the link between other-shock and

self-freezing activities then the decoding performance of behavioral clusters and freezing behavior should be affected. The authors should do this analysis here.

Response:

The decoding performance is shown in **Supplementary Fig. 11b,c,e,f** (previous Supplementary Fig. 7b,c,e,f). The decoding performance for freezing remained significant in both experiments. This suggests that the information on freezing and shock is independent at the single-cell level. Also, the presence of freezing-correlated neurons was unaffected, suggesting that the single information selectivity remains intact even if the mixed selectivity (i.e., shock information) is lost.

6/ This is a really interesting sets of experiments. However, the authors did not use the same analytic approach described in figure 1 and they quantified escape behavior with a different strategy than in figure 2i. This lack of consistency in the methods across the manuscript do not allow the reader to trust the results and the proposed model in figure 6j.

Response:

As the reviewer mentioned, to use the same analytic approach described in previous figures, we newly applied the same 2-s analysis in **Fig.7g,h**. The proportion of i_1 "escaped" was larger, and that of i_2+i_3 "gazing" was smaller in the inhibition group compared to the control group, but these differences were not significantly different. However, the difference was able to be depicted using the near/far analysis, so we kept the near/far analysis here. These results together suggest that the effects of the circuits are small, which is in line with the small effects at the neural representation level. The inhibition of input from those regions did not change the neural representation of other-shock and self-freezing themselves since both types of neurons were still present, but rather they affected the mixed representation at the single-cell and subpopulation levels.

We changed the sentence below in the **Result** section (p.10, line 277-279).

"While the proportion of i_1 "escaped" tends to be larger and i_2+i_3 "gazing" smaller in the inhibition group compared to the control group, these differences did not reach statistical significance (Fig. 7g,h). To describe the difference in detail, ..."

Reviewer #3:

In this manuscript the authors investigate the neural representation in the vmPFC of a mouse self-state related to a conspecific receiving repetitive foot shocks, using an observational fear paradigm. In this task, they made an interesting classification of stereotypic behavioural pattern thus increasing the number of observations usually limited, in the current literature, to measurement of freezing behavior. Building on previous reports showing the involvement of brain regions such as the ACC and the BLA in the acquisition of vicarious fear, the authors made an optogenetic study extending the role of these regions in a circuit that involves the vmPFC. Specifically the authors show:

They made a noteworthy behavioral classification, presented in Figure 1, showing that this approach can detect other behaviors than freezing, such as escape (cluster VIII). However, in Figure 6, to show the role of ACC-vmPFC and BLA-vmPFC circuits in freezing vs. escape behavior, the authors used a different behavioral analyses (far vs. near side of the cage). Thus is not clear why the authors elaborated such sophisticated behavioural analysis.

Also related to the behavioral analysis, it not clear how 'back' (cluster III) is different from 'escape' (cluster VIII). A mouse avoiding gazing his conspecific could also be considered an escape behavior. Indeed, in figure 2 it seems that optogenetic inhibition of the vmPFC also reduced cluster III.

Response:

We thank the reviewer for the thorough evaluation of our manuscript and constructive discussion. We have tried to address all the reviewer's comments in the revised manuscript.

For the behavioral classification, we developed a 10-s bout analysis in the previous submission. In this revised version, we added a new 2-s component analysis to more rigorously identify the OF-induced behaviors, as suggested by reviewer 2 (**Fig. 2**). We were able to extract OF-induced i1 "escaped" and i2 & i3 "gazing" behaviors using this analysis (**Fig. 2c-e**). With the vmPFC inhibition, we found the proportion of the i1 escaped behavior decreased and i2+i3 gazing behavior increased (**Fig. 3h-j**). In the ACC-vmPFC and BLA-vmPFC inhibition, though not statistically significant, the proportion changed in the inverse direction: an increase in i1 escaped and a decrease in i2+i3 gazing components (**Fig. 7g,h**). Moreover, we were able to identify neurons that were specifically active during i1 and i2+i3 (**Fig. 4j**).

In addition, this new analysis at a fine time scale revealed, as the reviewer pointed out, that the behavior component i1 "escaped", commonly appeared in clusters III and VIII as a large portion (**Fig. 2f**). The proportion of this component was indeed reduced by vmPFC inhibition (**Fig. 3i**), strengthening our previous conclusion of the involvement of vmPFC in escape behavior provided by 8 cluster classification (**Fig. 3g**).

We added sentences regarding the new 2-s analysis in the **Result** section.

(p.6, line 112-124)

“We conducted a similar analysis using shorter 2-s bouts to isolate the vicarious fear responses further. As mice move and stop continuously, we set a threshold of 50 % in the 2-s freezing rate to classify bouts as mobile or immobile. We then performed unsupervised classification and obtained 9 components for each dataset (Fig. 2a,b, Supplementary Fig. 4a,b). Among all the components (Supplementary Fig. 4c,d), three characteristic components were extracted from the mobile dataset, namely (m1) approaching, (m2) leaving, (m3) moving near D, three from the immobile dataset, (i1) escaped, (i2) gazing-1, (i3) gazing-2 (Fig. 2c). The freezing rate of m1 and m2 were significantly lower compared to other mobile components (Fig. 2d, Supplementary Table). When the habituation and conditioning period were compared, the proportion of mobile components (m1–m3) decreased, while the immobile components (i1–i3) increased (Fig. 2e). Among the eight clusters obtained in Fig. 1, m1 shared the largest portion of cluster I, m2 of cluster II, m3 of cluster I and VII, i1 of cluster III and VIII, i2 of V, and i3 of VII (Fig. 2f) indicating that the 10-s clusters contained a mixture of different behavior components.”

(p.6 line 139-141).

“In line with this, the proportion of the i1 “escaped” was significantly smaller and i2+i3 “gazing” was significantly larger in the inhibition group compared to the control group (Fig. 3h–j). To further confirm this trend, ...”

(p.7, line 168-173)

“We further investigated the 2-s behavior component information encoding in the vmPFC. Since two components, i1 and i2+i3, showed significant decrease and increase, respectively, in the vmPFC inhibition group compared to the control group (Fig. 3), we hypothesized these information is particularly represented in vmPFC neurons. We found 17 of 355 neurons significantly active at the component i1 and 11 of 355 at the component i2+i3 with 2 overlapping neurons (Fig. 4j).”

(p.10, line 277-279).

“While the proportion of i1 “escaped” tends to be larger and i2+i3 “gazing” smaller in the inhibition group compared to the control group, these differences did not reach statistical significance (Fig. 7g,h). To describe the difference in detail, ...”

The identified behavioral clusters are then used to study the neurophysiology related to these events. This is very interesting and the authors showed that vmPFC activity is also specific for the behavioral cluster. However, figure 3 showed that only 7 cells were activated by individual clusters. The authors used n=4 mice for this analysis, thus, this means that even less cells were activated for each animal? and in some animals no cells were activated for the clusters? Thus it is difficult to assess how relevant is the activation of these cells for the subsequent measurements in the manuscript.

Response:

In the previous analysis, we only identified 7 cells for 4 mice. However, with the suggestion from reviewer 2, we reconsidered the criteria of the cluster-specific neurons. Previously we used $\alpha = 0.025$ (two-tailed) with Bonferroni correction, but in the revised manuscript, we used $\alpha = 0.05$ with Holm-Bonferroni correction (changed in the **Methods** as shown below). We changed the alpha because it is difficult to identify cluster-specifically suppressed neurons with the firing data inferred from the calcium activity. In this way, 41 of 355 neurons were identified as cluster-specific neurons (11.5%, **Fig. 4e**). Regarding behavior components, we identified i1 component-specific neurons and i2+i3 component-specific neurons (**Fig. 4j**, we added this part in the **Methods** as shown below). Therefore, we consider our behavior analysis revealed that there are OF-induced self-behavior representations in the vmPFC.

We changed the **Result** section as follows (p.7, line 156-158).

“Consequently, we found that 11.5 % of cells (41 of 355) were significantly activated in individual clusters (Fig. 4e), while 9.5 % of cells (36 of 377) were identified as cluster-specific neurons in the no-shock control group (Supplementary Fig. 6d,e).”

We changed the **Methods** section as follows (p.21, line 600-606).

“Identification of cluster-specific and component-specific neurons

We used a permutation test to identify the cluster-specific and component-specific neurons. We circularly shifted the timing of calcium events using a random number for each mouse 10,000 times and calculated the sum of the calcium events for each cluster and component. We calculated the p-value of each cell for each cluster or i1 and i2+i3 components if the number of each cluster or component is more than 1 % of all bouts, and after Holm-Bonferroni correction ($p < 0.05$, one-sided), named significant cells as cluster-specific or component-specific neurons.”

In the following paragraph the authors described shock-activated and -suppressed cells and freezing-positive and -negative cells in the vmPFC and in the circuits including the ACC and the BLA. It is not clear, how the authors reconcile the observation of self-freezing cells in the vmPFC with the fact that perturbation of this area do not affect freezing behavior.

Response:

Indeed our optogenetic inhibition of the vmPFC, as well as the ACC-vmPFC and BLA-vmPFC, did not alter the freezing rate (**Fig. 3c, 7c, f**). We consider there are two potential explanations for this result. Firstly, it is possible that the vmPFC does not play a role in the output of freezing behavior. Alternatively, since our data revealed the presence of both freezing positively- and negatively-correlated neurons, it is possible that these subpopulations counteracted each other, leading to no observable changes in the freezing rate.

Moreover, after the initial description of avoidance behavior, the authors then jump to other-shock and self-freezing encoding in the vmPFC and is not clear how this is linked to the escape behavior and to the activity of the vmPFC cells during the behavioral clusters.

Response:

Thank you for the comment. We agree that our initial manuscript did not provide enough information regarding this point. While we referred to the behavior of the demonstrator mice as periodic, the behavior of the observer mice was also periodic. We added this data to show that the observer's behavior is related to the shock toward the demonstrator in both freezing and 2-s behavior component levels (**Supplementary Fig. 8a,c**). Therefore, we hypothesized that other-shock information might affect the escape behavior information in the vmPFC. To test this, we investigated whether other-shock information is represented in the vmPFC, and if it is, how the information is mixedly represented with self-related information, especially self-behaviors. In the previous manuscript, we only showed the relationship between shock and freezing. With the new 2-s bouts analysis, we also identified i1 'escaped' neurons (**Fig. 4j**). Those neurons were the main component of the shock-correlated and freezing-correlated neurons and showed a significantly high correlation with freezing and a low trend with shock (**Supplementary Fig. 10**).

We added sentences regarding the periodic behavior of observer mice and the neural property of component neurons in the **Result** section.

(p.8, line188-193)

"We observed periodic features in the observer's behavioral changes during each shock frequency (10 s) (Supplementary Fig. 8). Although freezing rate was significantly lower at the shock timing (0–2 s, Supplementary Fig. 8a), the average movement speed along the x-axis had no temporal feature (Supplementary Fig. 8b). These behavioral features indicate that the observer's behavior is dependent on demonstrator being shocked. Thus, to..."

(p.9, line 219-227)

"Furthermore, we investigated the neural properties of i1 "escaped"-specific and i2+i3 "gazing"-specific neurons, which are major components of the freezing-positively correlated neurons and shock-suppressed neurons (Supplementary Fig. 10a–d). The calcium activity of these neurons was suppressed during the shock timing (Supplementary Fig. 10e,f), and their correlation with the freezing rate was significantly higher than that of non-significant neurons (Supplementary Fig. 10g,i). Also, while not significant, shock correlation showed a lower trend than non-significant neurons (Supplementary Fig. 10h,j). Notably, we confirmed that the activity of the i1-specific neurons was high at the timing of component i1 compared to the preceding and proceeding 2-s bouts and component i2+i3 compared to the proceeding 2-s bouts (Supplementary Fig. 10k,l)."

Other comments related to the manuscript:

- Observers' freezing behavior during habituation seems significantly increased compared to the demonstrator? are the demonstrators familiar? do the observers received previous exposure to the shocks?

Response:

Observers indeed showed increased freezing behaviors during the habituation period. We consider this is due to their memory of having received a single weak shock (2s, 0.75 mA) 24 hours prior to the OF task (pre-shock), though in a different context (scent, floor, and left/right side of the observational chamber). The observers were co-housed with non-littermate demonstrators for seven consecutive days preceding the OF task. The detailed procedure is described in the **Methods** section (p.17, line 460-480).

- Why do the authors used different shock intensities across experiments?

Response:

It is known that the shock intensity affects the level of observational fear (Terranova et al, 2022), and 1.0 mA is typically used as strong shocks, while 0.5 mA is used as weak shocks. In our shock-delivery system, both demonstrator and observer mice showed high freezing rates in 0.75 mA and 1.00 mA conditions without any significant differences (**Supplementary Fig. 1d**). We mentioned the experiment in the **Methods** section (p.17 , line 480-481).

“We did not observe any behavioral difference between groups using 0.75 mA and 1.0 mA in our setting (Supplementary Fig. 1d).”

- In Fig. 2, for the optogenetic inhibition the authors applied 13 mW green light during the OF task, while 5mW for in vivo single unit recording. Why do the authors used different light intensities?

Response:

The intensity difference is mainly from the different experimental conditions; unilateral or bilateral. Optogenetic inhibition during the OF task was applied to the bilateral vmPFC (**Fig. 3a**), while the in vivo single-unit recording was performed unilaterally (**Fig. 3b**). Additionally, the light intensities were measured at the tip of each patch cable that was directly connected to fiber-optic cannula; a dual fiber-optic cannula (Ø200 µm core) for the OF task and a mono fiber-optic cannula (Ø105 µm core) for the single unit recording. We also considered the diameter of the fiber and the distance between the fiber tip and the target coordinate for adjustment. This is the reason why we used approximately half of the laser intensity for the single-unit recording.

We added a more detailed description of this in the **Methods** section (p.16, line 434-435).

“... optogenetic inhibition (5 mW at the fiber tip; approximately a half of laser intensity used in the OF task due to unilateral inhibition) was delivered to the recording site.”

REVIEWERS' COMMENTS

Reviewer #1 (Remarks to the Author):

The authors have adequately addressed my concerns, and have gone to some lengths to address the concerns of the other reviewers.

Reviewer #2 (Remarks to the Author):

The authors have conducted a very thorough revision of the original manuscript which addresses all of my comments. In particular, the addition of new behavioral and neuronal analysis provides a more holistic comprehension of the role of vmPFC in encoding vicarious fear responses. I now recommend the paper for publication.

Reviewer #3 (Remarks to the Author):

In this revised version of the manuscript the authors considered all the input of the reviewers, they included additional analysis and improved the presentation of the results and the discussion. I believe this study will be of interest to the readers of Nature Communications

Point-by-point responses:

We thank the reviewers for their suggestions and feedback, which have helped significantly improve the paper.

Reviewer #1:

The authors have adequately addressed my concerns, and have gone to some lengths to address the concerns of the other reviewers.

Response:

We thank the reviewer for the supportive comments.

Reviewer #2:

The authors have conducted a very thorough revision of the original manuscript which addresses all of my comments. In particular, the addition of new behavioral and neuronal analysis provides a more holistic comprehension of the role of vmPFC in encoding vicarious fear responses. I now recommend the paper for publication

Response:

We thank the reviewer for the constructive suggestions throughout the peer review process.

Reviewer #3:

In this revised version of the manuscript the authors considered all the input of the reviewers, they included additional analysis and improved the presentation of the results and the discussion. I believe this study will be of interest to the readers of Nature Communications

Response:

We thank the reviewer for the valuable input.